# SwiftTS: A Swift Selection Framework for Time Series Pre-trained Models via Multi-task Meta-Learning

**Tengxue Zhang**[1], **Biao Ouyang**[1], **Yang Shu**[1*], **Xinyang Chen**[2*], **Chenjuan Guo**[1], **Bin Yang**[1]

[1]East China Normal University, [2]Harbin Institute of Technology (Shenzhen)

{txzhang,bouyang}@stu.ecnu.edu.cn,chenxinyang@hit.edu.cn
{yshu,cjguo,byang}@dase.ecnu.edu.cn

## Abstract

Pre-trained models exhibit strong generalization to various downstream tasks. However, given the numerous models available in a model hub, identifying the most suitable one by individually fine-tuning is time-consuming. In this paper, we propose **SwiftTS**, a swift selection framework for time series pre-trained models. To avoid expensive forward propagation through all candidates, SwiftTS adopts a learning-guided approach that leverages historical dataset-model performance pairs across diverse horizons to predict model performance on unseen datasets. It employs a lightweight dual-encoder architecture that embeds time series and candidate models with rich characteristics, computing patchwise compatibility scores between data and model embeddings for efficient selection. To further enhance the generalization across datasets, we introduce a horizon-adaptive expert composition module that dynamically adjusts expert weights, and the transferable cross-task learning with cross-dataset and cross-horizon task sampling to enhance out-of-distribution (OOD) robustness. Extensive experiments on 14 downstream datasets and 8 pre-trained models demonstrate that SwiftTS achieves state-of-the-art performance in time series pre-trained model selection. The code and datasets are available at https://github.com/decisionintelligence/SwiftTS.

## 1 Introduction

Time series forecasting (Cirstea et al., 2022; Cheng et al., 2023; Pan et al., 2023; Chen et al., 2024) is a fundamental task with broad applications in finance, weather prediction, and transportation (Guo et al., 2014; Ma et al., 2014; Yang et al., 2023; Tian et al., 2026; 2025; Mei et al., 2025; Feng et al., 2025; Wang et al., 2026). Inspired by the success of pre-trained models in natural language processing (Hurst et al., 2024; Yang et al., 2025) and computer vision (Dosovitskiy et al., 2021), numerous time series foundation models have been developed (Das et al., 2024; Gao et al., 2024; Wang et al., 2025; Wu et al., 2026). Pre-trained on large and diverse datasets, these models acquire transferable knowledge that can be adapted to downstream tasks through fine-tuning, eliminating the need for training from scratch (Kumar et al., 2022; Jia et al., 2022; Dettmers et al., 2023).

However, no single pre-trained model excels in all tasks (Li et al., 2025), making model selection for time series forecasting challenging. Although fine-tuning all candidate models provides model selection results, it is often computationally infeasible for large model pools. Therefore, developing efficient methods to identify the optimal pre-trained model is crucial for real-world deployment. Existing approaches, primarily designed for image models, are feature-analytic methods that analyze features extracted from the target dataset using pre-trained models: some assess feature-task alignment via statistical metrics (Nguyen et al., 2020), while others investigate intrinsic properties of the feature space (Pándy et al., 2022; Wang et al., 2023). Only a few are learning-based (Zhang et al., 2023), learning similarity functions between data and model representations for model selection. Despite recent advances, several challenges remain unresolved for time series pre-trained models:

---

*Corresponding authors

**Challenge 1: Oversight of model heterogeneity and time series data characteristics**. Current time series pre-trained models are typically heterogeneous in both architecture and training objectives, unlike the standardized feature extractors in vision models. This diversity hinders unified feature extraction and limits the applicability of many existing methods. Moreover, extracting features also requires costly forward passes through each model, leading to substantial computational overhead as the model hub or datasets scale. Some learning-based methods attempt to mitigate this cost via a shared feature extractor, but often sacrifice performance. In addition, time series data exhibit temporal dependencies and sequential patterns that are critical for accurate forecasting but largely ignored in current approaches. Valuable prior knowledge and intuitive insights, such as "models generally perform better when the downstream domain aligns with the pre-training domain," are also rarely incorporated into current model selection criteria.

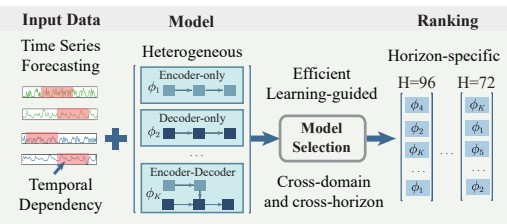

Figure 1: SwiftTS employs an efficient learning-guided selection framework for time series forecasting, enabling horizon-specific selection and improved cross-domain generalization via multi-task meta-learning.

**Challenge 2: Limited generalization in cross-domain and cross-horizon scenarios**. Feature-analytic methods often fail when the target dataset diverges from the pre-training domain. In such cases, fine-tuning can induce substantial shifts in model parameters, making initial features poor predictors of post-tuning performance. Although a few learning-based methods assume that the learning paradigm can generalize, they lack explicit designs to ensure robustness on unseen datasets. Moreover, the performance of time series pre-trained models can vary considerably across different forecasting horizons. A model that excels at short-term predictions may degrade on longer horizons. Ignoring such variability can result in misleading performance rankings that fail to reflect the dynamic, horizon-dependent nature of time series forecasting. Current methods often lack mechanisms to effectively select models tailored to specific forecasting horizons, as in Figure 1, thereby limiting their generalization across varying forecasting ranges.

To address **Challenge 1**, we propose an efficient learning-guided selection framework that avoids inconsistent feature extraction and reduces forward propagation costs. We collect the performance of various dataset-model pairs across different horizons as training data to learn how models perform on unseen datasets. The framework is tailored to time series characteristics and incorporates prior knowledge to enhance selection. The data encoder segments the input series into patches and generates patch-level embeddings that capture local temporal patterns and preserve sequential dependencies. The model encoder incorporates meta-information, topological structure, and functionality to construct a comprehensive representation of candidate models. By employing a lightweight dual-encoder architecture, our framework independently learns informative embeddings for both downstream datasets and candidate models. Finally, patch-level cross-attention assesses the fine-grained compatibility score between them, enabling accurate and efficient model selection.

To address **Challenge 2**, we propose a generalizable multi-task meta-learning strategy. To equip our framework with the multi-task flexibility to accommodate varying horizons, we incorporate a horizon-adaptive expert composition. This design dynamically assigns adaptive weights to multiple experts based on the target forecasting horizon, enabling horizon-specific ranking predictions. To further enhance generalization across both domains and horizons, we propose transferable cross-task learning to improve robustness in out-of-distribution (OOD) scenarios. We introduce a meta-learning paradigm with two task sampling strategies: (1) cross-dataset sampling, where tasks are drawn from different datasets to encourage inter-domain generalization, and (2) cross-horizon sampling, where tasks are constructed using varying forecasting horizons to improve horizon-level adaptability. By meta-learning from tasks across diverse datasets and horizons, our framework learns transferable knowledge that captures both task-specific characteristics and shared cross-task patterns. This ultimately improves its performance in real-world applications.

To the best of our knowledge, this is the first model selection method for time series pre-trained models. The contributions are summarized as follows:

- We propose a swift learning-guided framework that leverages a dual-encoder to embed datasets and models, computing patchwise compatibility scores for model selection.

- We introduce a multi-task meta-learning strategy with a horizon-adaptive expert composition to enhance generalization across datasets and forecasting horizons.

- Extensive experiments on benchmarks comprising 14 real-world downstream datasets and 8 pre-trained models show that our method achieves state-of-the-art performance in pre-trained model selection for time series forecasting.

## 2 RELATED WORK

**Time series Pre-trained Model.** Existing models can be broadly categorized into three main architectures: (1) *Encoder-only models*: MOIRAI (Woo et al., 2024) flattens multivariate sequences into unified sequences for Transformer pre-training, Moment (Goswami et al., 2024) employs masked reconstruction to train a versatile Transformer, and UniTS (Gao et al., 2024) introduces task tokenization and dynamic self-attention across temporal and variable dimensions. (2) *Decoder-only models*: TimesFM (Das et al., 2024) and Timer (Liu et al., 2024) adopt GPT-style designs for next-token prediction, achieving strong zero-shot performance. (3) *Encoder-decoder models*: TTM (Ekambaram et al., 2024) leverages MLP-Mixer blocks with multi-resolution sampling to capture cross-channel patterns. ROSE (Wang et al., 2024) enhances generalization through decomposed frequency learning and time series register components. Chronos (Ansari et al., 2024) adapts the T5 (Raffel et al., 2020) language foundation model to time series by discretizing data via binning and scaling. As the number and variety of time series pre-trained models continue to grow, efficiently and accurately selecting the most suitable model from a diverse model hub remains a challenge.

**Pre-trained Model Selection.** Pre-trained model selection aims to quickly identify the best model for downstream tasks from a model hub. Existing methods fall into two broad categories: (1) *Feature-analytic methods* analyze features extracted by pre-trained models from the target dataset. Early approaches, such as NCE (Tran et al., 2019) and LEEP (Nguyen et al., 2020), leverage statistical metrics but depend on pre-trained classifiers, limiting their applicability to self-supervised models. LogME (You et al., 2021) overcomes this by estimating the maximum label marginalized likelihood. RankME (Garrido et al., 2023) posits that models with higher feature matrix ranks exhibit superior transferability. Other methods focus on class separability during the fine-tuning process. GBC (Pándy et al., 2022) measures the degree of overlap between pairwise target classes based on extracted features. SFDA (Shao et al., 2022) enhances class separability by projecting features into a Fisher space. Etran (Gholami et al., 2023) introduces an energy-based transferability metric, while DISCO (Zhang et al., 2025) proposes a framework for evaluating pre-trained models based on the distribution of spectral components. (2) *Learning-based methods* aim to predict model transferability through a learning framework (Wu et al., 2024a;b). Model Spider (Zhang et al., 2023) learns model representations and a similarity function by aligning them with downstream task representations, enabling model selection via the learned similarity. Despite the diversity of existing methods, they often rely on costly feature extraction and generalize poorly across domains and forecasting horizons. To address these issues, we propose SwiftTS, a swift model selection framework via multi-task meta-learning. It infuses prior knowledge and adopts a learning-guided paradigm, avoiding expensive feature analysis used in prior work while improving OOD robustness.

## 3 PROBLEM FORMULATION

Given a model hub $Z = \{\phi_k\}_{k=1}^K$ of $K$ time series pre-trained models and a target dataset $D = \{x_i, y_i\}_{i=1}^N$ with $N$ samples, the goal is to select the model that achieves the best performance on the time series forecasting task with horizon $H$. Brute-force fine-tuning of all models yields the ground-truth performances $\{r_k\}_{k=1}^K$ for the model hub, but at prohibitive computational cost. To avoid this, model selection methods estimate transferability without fine-tuning by assigning each model $\phi_k$ an assessment score $\hat{r}_k$, where a larger $\hat{r}_k$ indicates stronger expected performance:

$$\hat{r}_k = f(\phi_k, D, H) \tag{1}$$

The $f$ is a scoring function that measures the compatibility between the model $\phi_k$ and the target dataset $D$ under the forecasting horizon $H$. Ideally, the predicted scores $\{\hat{r}_k\}_{k=1}^K$ should strongly correlate with the fine-tuning results $\{r_k\}_{k=1}^K$, enabling the selection of the most transferable model.

## 4 METHODS

We propose an efficient framework for pre-trained model selection in time series forecasting via multi-task meta-learning (Figure 2). The **learning-guided selection framework** adopts a dual-encoder architecture: a temporal-aware data encoder captures sequential patterns by segmenting time series into patches to generate data embeddings, while a knowledge-infused model encoder incorporates prior knowledge about models to construct rich model embeddings. Then, we employ patch-level cross-attention to evaluate fine-grained compatibility scores between them. To facilitate multi-task forecasting and enhance generalization, we further adopt a **generalizable multi-task meta-learning** strategy: a horizon-adaptive expert composition module adaptively assigns weights to experts based on the target horizon, and the transferable cross-task learning with cross-dataset and cross-horizon task sampling improves robustness under OOD scenarios. The framework is trained on a meta-dataset of $N$ samples, $\mathcal{D}_{\text{meta}} = \{D^i, Z, H^i, \boldsymbol{r}^i\}_{i=1}^N$, where the $i$-th sample includes a downstream dataset $D^i$, a shared model hub $Z$, the horizon $H^i$ and the corresponding ranking scores $\boldsymbol{r}^i$ of the model hub. The ranking scores reflect the relative performance of models in $Z$ on dataset $D^i$ under horizon $H^i$. This meta-dataset allows the framework to learn how models perform on various datasets, enabling accurate performance prediction and model selection on unseen datasets.

### 4.1 LEARNING-GUIDED SELECTION FRAMEWORK

**Temporal-Aware Data Encoder.** In time series modeling, capturing temporal dependencies and sequential patterns is essential for accurate forecasting. Relying solely on meta-information of time series makes it difficult to represent the fine-grained temporal structure. To effectively model temporal characteristics, we follow a well-established design and divide the time series $X \in \mathbb{R}^{L \times C}$ with $L$ time steps across $C$ variates into patches of size $S$, yielding $P = \lfloor L/S \rfloor$ patches. Each patch $X_p \in \mathbb{R}^{S \times C}$ is linearly projected into a $d$-dimensional embedding $X_p' \in \mathbb{R}^{1 \times d}$, forming patch embeddings $E_{\text{patch}} \in \mathbb{R}^{P \times d}$. To preserve temporal order, the positional encodings $E_{\text{pos}} \in \mathbb{R}^{P \times d}$ following (Vaswani et al., 2017) are added: $E_{\text{inp}} = E_{\text{patch}} + E_{\text{pos}}$.

The resulting embeddings $E_{\text{inp}}$ are fed into a self-attention module to capture long-range dependencies, where $W_Q^{sa} \in \mathbb{R}^{d \times d}$, $W_K^{sa} \in \mathbb{R}^{d \times d}$, and $W_V^{sa} \in \mathbb{R}^{d \times d}$ are the learnable projection matrices:

$$E_{\text{sa}} = \text{SA}(E_{\text{inp}}) = \text{softmax}\left( E_{\text{inp}} W_Q^{sa} \left(E_{\text{inp}} W_K^{sa}\right)^T / \sqrt{d_k} \right) E_{\text{inp}} W_V^{sa} \tag{2}$$

Downstream datasets are often large, making full-dataset encoding computationally expensive and limiting the diversity of learned representations. To address this, we introduce a multiple-subset sampling strategy, where multiple subsets of $B$ time series are sampled from the same dataset, and their attention outputs $E_{\text{sa}}$ are aggregated into $E_{\text{sub}} \in \mathbb{R}^{B \times P \times d}$. These subsets collectively offer a more comprehensive representation of the overall dataset. Aggregating the compatibility across these diverse subsets, rather than relying on a single random subset, more faithfully reflects the dataset-level compatibility and effectively balances intra-dataset variance. We then apply mean pooling along the batch dimension to produce a compact and informative data embedding $E_d \in \mathbb{R}^{P \times d}$, which captures shared temporal patterns within the subset while remaining robust to sample-level variability. Finally, the resulting $E_d$ serves as an expressive summary of downstream tasks, allowing for compatibility scoring with model embeddings during model selection.

**Knowledge-Infused Model Encoder.** Directly embedding a model with millions of parameters and a complex structure is challenging. To characterize a pre-trained model $\phi_k$, we infuse three key components: meta-information, topological structure, and functionality. This design aligns with human intuition in model selection (e.g., "choose a larger model for complex tasks"), which has been largely overlooked in existing methods. The **meta-information** embedding $v_a^k \in \mathbb{R}^{1 \times d_a}$ encodes prior knowledge from the pre-training to guide model selection. We consider: (1) Model architecture: Categorized into *encoder-only*, *decoder-only*, and *encoder-decoder*, reflecting structural design and training characteristics. (2) Model capacity: Estimated by parameter count, indicating ability to capture complex patterns. (3) Model complexity: Measured in Giga Multiply-Accumulate operations (GMACs). Generally, higher complexity allows the model to capture richer patterns. (4) Model dimension: The hidden dimension size across inputs, states, and outputs. Larger dimensions enable the model to learn more expressive and detailed information. (5) Pre-trained domain: Models pre-trained on similar domains typically transfer better. More details of these five types of meta-information for each model are provided in Appendix A.5.

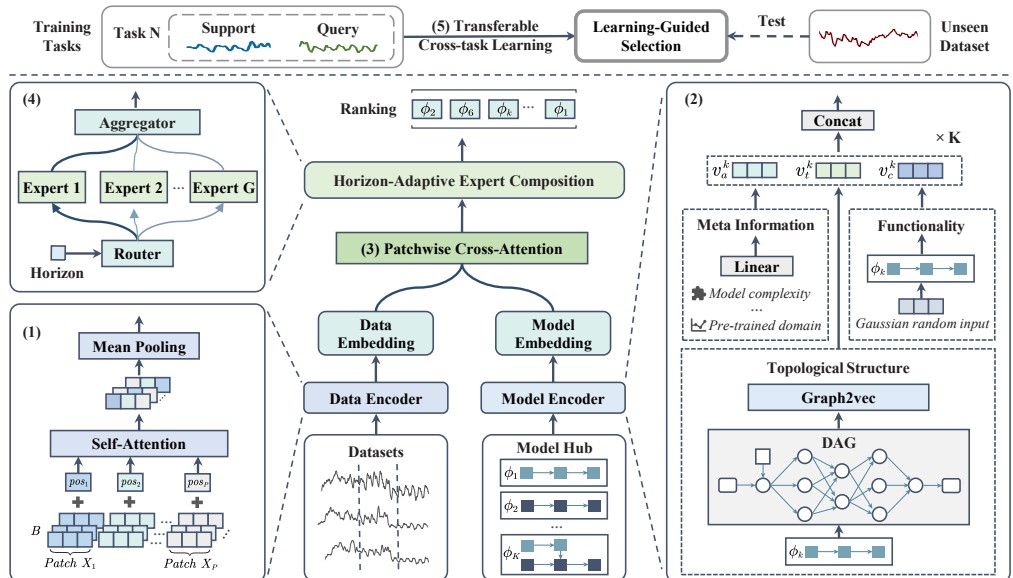

Figure 2: The framework of SwiftTS, consisting of (1) a temporal-aware data encoder, (2) a knowledge-infused model encoder, (3) patchwise cross-attention, (4) a horizon-adaptive expert composition module, and (5) the transferable cross-task learning.

The **topological structure** provides a detailed view of a model's architecture and inductive biases. Structural information, such as layer depth, data flow, and connectivity reveals how models process inputs, extract features, and make predictions. We first represent the architecture of the model $\phi_k$ as a directed acyclic graph (DAG), denoted by $G_k = (V_k, E_k, A_k)$, where $V_k$ and $E_k$ denote vertices and edges, and $A_k$ represents the node attributes. By leveraging the chain rule and gradient propagation maps, we trace how data flows through the network, thereby identifying the operations applied to the data and the directed paths along which the data travels. In this DAG, each node corresponds to a computational operation performed within the network (e.g., normalization, activation). The edges correspond to the directed paths along which the data are propagated, reflecting the computational dependencies and information flow within the model. Once the DAG is constructed, we employ graph2vec (Narayanan et al., 2017; Rozemberczki et al., 2020), an unsupervised graph embedding method inspired by doc2vec (Le & Mikolov, 2014) to obtain the topological embedding $\boldsymbol{v}_t^k \in \mathbb{R}^{1 \times d_t}$.

The **functionality** reflects how pre-trained parameters encapsulate the biases and knowledge acquired during pre-training. Since directly embedding millions of parameters is infeasible, we adopt a functional embedding inspired by model distillation, which characterizes a model through its input-output behavior. The intuition is simple: models with different architectures or parameters implement distinct functions and thus are expected to produce distinguishable outputs on identical inputs. Although using real or synthetic time-series inputs can reflect how a model responds to the real-world time series, it causes the resulting functional embedding to inadvertently inherit biases or domain-specific priors from the chosen probing data. For example, a model would yield a more favorable embedding than others simply because the probing inputs are similar to its pre-trained data, which is unfair to other models. In contrast, random Gaussian noise serves as a neutral stimulus, which enables us to observe the model's intrinsic input-output behavior without imposing external assumptions. Therefore, we feed a fixed set of Gaussian random noise inputs $\epsilon \sim \mathcal{N}(0, I)$ into each model $\phi_k$ and record the outputs $\boldsymbol{v}_c^k = \phi_k(\epsilon)$ as its functional embedding, where $\boldsymbol{v}_c^k \in \mathbb{R}^{1 \times d_c}$.

Finally, we integrate the meta-information embeddings $\boldsymbol{v}_a \in \mathbb{R}^{K \times d_a}$, the topological embeddings $\boldsymbol{v}_t \in \mathbb{R}^{K \times d_t}$, and the functional embeddings $\boldsymbol{v}_c \in \mathbb{R}^{K \times d_c}$ of the model hub by concatenation, then project the result through a linear transformation $W_m \in \mathbb{R}^{d \times (d_a + d_t + d_c)}$ and a nonlinear activation $\sigma$ to generate the final model embedding $\boldsymbol{E}_m \in \mathbb{R}^{K \times d}$:

$$\boldsymbol{E}_m = \sigma([\boldsymbol{v}_a, \boldsymbol{v}_t, \boldsymbol{v}_c]W_m^T) \tag{3}$$

**Patchwise Compatibility Score.** To facilitate a fine-grained and context-aware comparison between downstream datasets and pre-trained models, we compute a compatibility score using patchwise

cross-attention (CA). Unlike global similarity measures, patchwise cross-attention captures localized correspondences by assessing each data patch's contribution to overall compatibility. Specifically, the model embedding $E_m$ as the query, and the data embedding $E_d$ as the key and value:

$$E_{\text{ca}} = \text{CA}(E_m, E_d) = \text{softmax} \left( E_m W_Q^{ca} \left( E_d W_K^{ca} \right)^T / \sqrt{d_k} \right) E_d W_V^{ca} \tag{4}$$

where $W_Q^{ca} \in \mathbb{R}^{d \times d}$, $W_K^{ca} \in \mathbb{R}^{d \times d}$, $W_V^{ca} \in \mathbb{R}^{d \times d}$ are projection matrices. This mechanism enables the model to focus on semantically meaningful patches in the data that are most relevant to the characteristics of the model. Finally, a multi-layer perceptron (MLP) produces the ranking prediction, where $\hat{r} \in \mathbb{R}^K$ denotes the predicted ranking scores for $K$ candidate pre-trained models:

$$\hat{r} = \text{MLP}(E_{\text{ca}}) \tag{5}$$

**Learn-to-select Optimization.** During training, we adopt a joint objective combining ranking regularization and prediction accuracy. The ranking loss enforces correct relative orderings among pre-trained models, while the prediction loss (MSE) ensures precise performance estimation:

$$\mathcal{L}_{\text{total}} = \underbrace{-\sum_{k=1}^{K} p_k(\hat{r}) \log q_k(r)}_{\text{ranking loss}} + \lambda \cdot \underbrace{\sum_{k=1}^{K} \|r_k - \hat{r}_k\|_2^2}_{\text{prediction loss}} \tag{6}$$

where $\hat{r}$ and $r$ denote the predicted and ground-truth ranking scores, $p_k(\hat{r})$ and $q_k(r)$ are their softmax-normalized forms of the $k$-th model, and $\hat{r}_k$, $r_k$ are the corresponding individual scores.

## 4.2 GENERALIZABLE MULTI-TASK META-LEARNING

**Horizon-Adaptive Expert Composition.** Time series models often perform inconsistently across forecasting horizons, leading to varying rankings. To equip our framework with the multi-task flexibility to accommodate varying horizons, we propose a horizon-adaptive expert composition module that dynamically integrates specialized experts for different horizons. A lightweight router network assigns softmax-normalized weights to $G$ experts based on the target horizon $H$:

$$w = \text{softmax}(\text{Router}(H; \theta_s)) \tag{7}$$

where $\theta_s$ denotes the parameters of the router, and $w \in \mathbb{R}^G$ the expert weights. Each expert, implemented as an MLP, processes the cross-attention output $E_{\text{ca}}$ to generate the final prediction through a linear combination of the expert outputs to replace Equation (5):

$$\hat{r} = \sum_{g=1}^{G} w_g \cdot \text{MLP}_g(E_{\text{ca}}) \tag{8}$$

This design flexibly adapts to diverse horizons without the need for retraining, enhancing both parameter sharing across tasks and improving computational efficiency.

**Transferable Cross-Task Learning.** Existing methods often struggle to generalize when faced with datasets that deviate significantly from the pre-training distribution. This presents a major challenge in real-world applications, where data distribution can vary widely across different domains. This issue is particularly pronounced in time series forecasting, where model performance is not only influenced by the nature of the dataset but is also highly sensitive to the forecasting horizon. To achieve robust and flexible model ranking and selection, we focus on two types of OOD scenarios: (1) generalizing model rankings from previously seen datasets to unseen datasets, which requires learning transferable features that capture the essential characteristics governing model performance; and (2) transferring performance prediction across different horizons, which demands understanding how model capabilities evolve with forecasting horizons.

To enable transferable cross-task learning and enhance the model's ability to adapt to diverse scenarios, we incorporate meta-learning (Finn et al., 2017) into our framework. Given the constructed meta-dataset $\mathcal{D}_{\text{meta}} = \{D^i, Z, H^i, r^i\}_{i=1}^{N}$, we sample a set of diverse tasks $\mathcal{T} = \{\mathcal{T}_1, \mathcal{T}_2, ..., \mathcal{T}_n\}$, where each task is divided into a support set and a query set. During training, as shown in Figure 3, the support set is used to simulate fast adaptation to new conditions without updating the parameters,

referred to as the inner-loop update. Subsequently, the query set evaluates the model's performance after adaptation and involves actual parameter updates, constituting the outer-loop update.

We introduce two task sampling strategies to explicitly encourage effective cross-domain and cross-horizon generalization: (1) **cross-dataset sampling**, where tasks and their support and query sets are drawn from different datasets to promote generalization across domains. (2) **cross-horizon sampling**, where tasks and their support and query sets are constructed from varying forecasting horizons to enhance adaptability at the horizon level. By exposing the model to these heterogeneous conditions during meta-training, it is encouraged to identify both shared and domain-specific patterns, recognize the underlying structures that govern pre-trained model performance, and improve model selection across diverse forecasting scenarios.

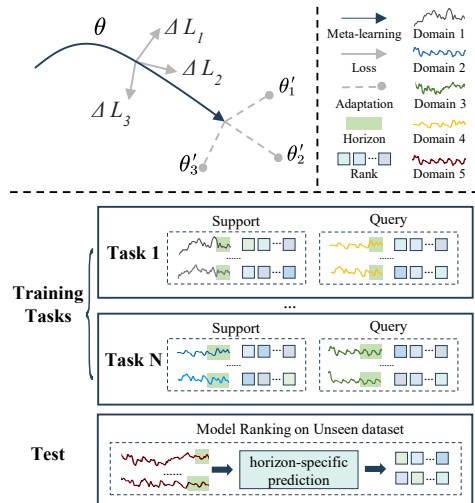

In practice, downstream datasets and target forecasting horizons often exhibit greater diversity. To simulate more realistic and challenging conditions, we combine the above two strategies into a unified sampling approach. Specifically, within a single meta-training task, the support set and the query set are randomly sampled from disjoint collections of datasets and forecasting horizons, ensuring that there is no distribution leakage and minimizing task redundancy.

To formalize the cross-task learning process, we treat each sampled task as an independent learning episode during training. In each episode, the model first adapts to a *support set*, simulating rapid adjustment to a new domain or forecasting horizon. The adapted parameters are then evaluated on a *query set*, and the meta-gradients are computed based on the performance of this query set. These gradients are subsequently used to update the initial model parameters in a direction that improves generalization across tasks.

Figure 3: Parameter update process in cross-task learning (top-left), and sampling strategies for cross-horizon and cross-dataset tasks (bottom).

Formally, let $\theta$ represent the initial parameters of the model. For a given task $\mathcal{T}_i$, we compute the task-specific parameters $\theta_i'$ via a few gradient steps on the support set:

$$\theta_i' = \theta - \alpha \nabla_\theta \mathcal{L}_{\text{supp}}(\mathcal{T}_i; \theta) \tag{9}$$

where $\alpha$ is the inner-loop learning rate and $\mathcal{L}_{\text{supp}}(\mathcal{T}_i; \theta)$ denotes the loss function evaluated on the support set of task $\mathcal{T}_i$, as defined in Equation (6). The adapted parameters $\theta_i'$ are then evaluated on the query set to compute the meta-objective that measures how well the model generalizes after adaptation: $\mathcal{L}_{\text{query}}(\mathcal{T}_i; \theta_i')$. This loss is also computed as defined in Equation (6). Finally, the model parameters $\theta$ are updated using the meta-gradient across all tasks:

$$\theta \leftarrow \theta - \gamma \nabla_\theta \sum_{\mathcal{T}_i} \mathcal{L}_{\text{query}}(\mathcal{T}_i; \theta_i'), \tag{10}$$

where $\gamma$ is the outer-loop learning rate. By repeatedly performing the two-step optimization process, which consists of inner-loop adaptation and outer-loop generalization, the model learns parameters that enable rapid adaptation to new domains or forecasting horizons with minimal data and computational resources. Please refer to Algorithm 1 for the overall algorithmic process.

# 5 EXPERIMENTS

## 5.1 EXPERIMENTAL DESIGN

**Datasets.** We evaluate SwiftTS on 14 public time series forecasting datasets across diverse domains, including electricity (ETTh1/ETTh2 (Zhou et al., 2021), ETTm1/ETTm2 (Zhou et al., 2021), Electricity (Trindade, 2015)), energy (Solar (Lai et al., 2018), Wind (Li et al., 2022)), traffic (PEMS08 (Song et al., 2020), Traffic (Wu et al., 2021)), environment (Weather (Wu et al., 2021), AQShunyi (Zhang et al., 2017)), natural (ZafNoo (Poyatos et al., 2021), CzeLan (Poyatos et al.,

Table 1: Method comparison of the weighted Kendall's $\tau_\omega$ across 14 datasets and their average. The best and second-best results are in bold and underlined. Our method achieves the best overall $\tau_\omega$.

| | Horizon | RankME | LogME | Regression | Etran | DISCO | AutoForecast | Model spider | zero-shot | SwiftTS |
|---|---|---|---|---|---|---|---|---|---|---|
| ETTh1 | H=96 | 0.292 | 0.257 | 0.257 | 0.287 | 0.213 | 0.243 | 0.309 | 0.091 | **0.464** |
| | H=192 | 0.182 | 0.014 | 0.104 | 0.147 | 0.059 | 0.233 | 0.386 | -0.058 | **0.479** |
| | H=336 | 0.161 | 0.137 | 0.120 | 0.158 | -0.036 | 0.232 | 0.321 | -0.126 | **0.436** |
| | H=720 | 0.086 | 0.196 | 0.108 | 0.147 | 0.019 | 0.329 | 0.334 | 0.241 | **0.543** |
| ETTh2 | H=96 | 0.116 | 0.089 | 0.078 | 0.256 | 0.156 | 0.296 | 0.398 | 0.211 | **0.436** |
| | H=192 | 0.101 | -0.150 | -0.059 | 0.333 | -0.165 | 0.197 | **0.451** | 0.430 | 0.330 |
| | H=336 | -0.011 | -0.134 | -0.083 | 0.269 | -0.154 | 0.185 | **0.389** | 0.256 | 0.219 |
| | H=720 | 0.053 | -0.014 | -0.069 | 0.106 | 0.188 | 0.246 | 0.345 | **0.474** | 0.351 |
| ETTm1 | H=96 | 0.005 | 0.126 | 0.295 | 0.691 | -0.407 | -0.243 | **0.734** | 0.063 | 0.501 |
| | H=192 | 0.005 | 0.079 | 0.079 | 0.552 | -0.406 | -0.243 | **0.567** | -0.040 | 0.501 |
| | H=336 | -0.529 | -0.048 | -0.048 | 0.099 | 0.191 | 0.075 | 0.590 | -0.276 | **0.652** |
| | H=720 | -0.479 | -0.027 | -0.027 | 0.173 | 0.166 | 0.169 | 0.266 | -0.041 | **0.269** |
| ETTm2 | H=96 | 0.140 | 0.080 | 0.334 | **0.913** | 0.613 | -0.164 | 0.667 | 0.285 | 0.847 |
| | H=192 | 0.128 | -0.020 | 0.164 | 0.568 | 0.520 | -0.243 | 0.696 | 0.319 | **1.000** |
| | H=336 | -0.118 | -0.021 | 0.159 | 0.700 | 0.473 | -0.304 | 0.698 | **0.727** | 0.663 |
| | H=720 | -0.068 | 0.050 | 0.163 | 0.485 | 0.165 | -0.155 | 0.356 | 0.078 | **0.534** |
| Electricity | H=96 | -0.037 | 0.317 | 0.317 | **0.559** | -0.026 | -0.060 | 0.311 | 0.162 | 0.361 |
| | H=192 | 0.043 | 0.306 | 0.306 | **0.685** | -0.585 | -0.161 | 0.549 | 0.093 | 0.240 |
| | H=336 | -0.330 | -0.419 | -0.419 | -0.009 | 0.091 | -0.098 | 0.358 | -0.279 | **0.520** |
| | H=720 | -0.403 | -0.344 | -0.344 | 0.027 | 0.148 | 0.009 | 0.356 | 0.029 | **0.394** |
| Traffic | H=96 | -0.138 | 0.059 | -0.124 | -0.096 | 0.148 | 0.114 | 0.049 | -0.312 | **0.247** |
| | H=192 | -0.444 | -0.432 | -0.432 | -0.338 | 0.225 | 0.324 | -0.005 | -0.198 | **0.351** |
| | H=336 | -0.569 | -0.568 | -0.568 | -0.692 | **0.643** | 0.092 | 0.030 | -0.026 | 0.066 |
| | H=720 | -0.833 | -0.745 | -0.565 | -0.394 | 0.223 | 0.092 | 0.420 | 0.222 | **0.702** |
| Solar | H=96 | -0.214 | 0.120 | 0.222 | 0.359 | -0.569 | -0.050 | **0.452** | 0.308 | 0.222 |
| | H=192 | -0.017 | **0.473** | 0.430 | 0.246 | -0.022 | -0.070 | 0.344 | 0.110 | 0.119 |
| | H=336 | -0.003 | 0.553 | **0.553** | -0.359 | -0.296 | -0.145 | 0.250 | -0.395 | 0.277 |
| | H=720 | -0.062 | 0.527 | **0.527** | 0.055 | -0.703 | -0.305 | 0.051 | -0.027 | 0.512 |
| Weather | H=96 | 0.015 | -0.381 | -0.034 | 0.499 | 0.273 | -0.274 | **0.543** | 0.128 | 0.285 |
| | H=192 | -0.125 | -0.023 | -0.118 | **0.466** | 0.024 | -0.130 | 0.257 | 0.341 | 0.129 |
| | H=336 | -0.171 | 0.102 | -0.066 | **0.539** | 0.404 | -0.193 | 0.340 | 0.321 | 0.108 |
| | H=720 | -0.065 | -0.057 | 0.241 | **0.745** | 0.375 | -0.155 | 0.251 | 0.702 | 0.316 |
| Exchange | H=96 | -0.032 | -0.492 | -0.343 | 0.193 | 0.112 | -0.598 | 0.059 | -0.284 | **0.251** |
| | H=192 | -0.040 | -0.597 | -0.414 | 0.246 | 0.152 | -0.404 | 0.154 | -0.273 | **0.252** |
| | H=336 | -0.112 | -0.536 | -0.306 | 0.148 | **0.250** | -0.552 | 0.230 | -0.143 | 0.233 |
| | H=720 | -0.210 | -0.617 | -0.444 | -0.024 | **0.536** | -0.733 | 0.322 | 0.126 | -0.386 |
| ZafNoo | H=96 | -0.117 | 0.058 | -0.148 | 0.113 | -0.153 | 0.454 | 0.511 | -0.384 | **0.656** |
| | H=192 | -0.285 | -0.235 | -0.274 | -0.044 | -0.035 | 0.454 | 0.436 | -0.017 | **0.786** |
| | H=336 | -0.285 | 0.040 | -0.133 | -0.073 | -0.200 | 0.454 | 0.580 | 0.023 | **0.732** |
| | H=720 | -0.224 | -0.106 | -0.256 | 0.129 | -0.235 | 0.478 | 0.552 | 0.164 | **0.668** |
| CzeLan | H=96 | 0.103 | 0.071 | 0.012 | **0.632** | -0.343 | 0.574 | -0.121 | 0.171 | 0.575 |
| | H=192 | -0.037 | -0.376 | -0.258 | 0.337 | 0.362 | 0.499 | 0.217 | 0.171 | **0.527** |
| | H=336 | -0.069 | -0.090 | -0.155 | -0.109 | -0.154 | 0.573 | -0.009 | 0.301 | **0.847** |
| | H=720 | -0.125 | -0.214 | -0.074 | -0.125 | -0.136 | 0.519 | 0.349 | 0.282 | **0.839** |
| AQShunyi | H=96 | -0.371 | -0.283 | -0.270 | -0.117 | 0.107 | 0.390 | 0.414 | -0.349 | **0.939** |
| | H=192 | -0.407 | 0.328 | 0.328 | -0.045 | -0.224 | 0.438 | 0.126 | 0.309 | **0.734** |
| | H=336 | -0.377 | -0.184 | -0.255 | -0.277 | 0.276 | 0.411 | 0.084 | 0.420 | **0.723** |
| | H=720 | -0.332 | -0.140 | -0.209 | -0.253 | 0.119 | 0.370 | 0.335 | 0.675 | **0.701** |
| Wind | H=96 | 0.211 | 0.142 | 0.062 | **0.417** | -0.196 | 0.251 | 0.244 | -0.106 | 0.395 |
| | H=192 | 0.211 | 0.258 | 0.231 | 0.136 | **0.482** | 0.251 | 0.355 | -0.155 | 0.395 |
| | H=336 | 0.045 | **0.545** | 0.319 | 0.338 | -0.532 | 0.097 | 0.281 | 0.349 | 0.262 |
| | H=720 | -0.040 | **0.474** | 0.443 | 0.126 | -0.377 | -0.015 | 0.202 | 0.133 | 0.162 |
| PEMS08 | H=96 | 0.140 | 0.118 | 0.118 | -0.253 | 0.110 | 0.036 | -0.103 | **0.445** | 0.401 |
| | H=192 | 0.038 | -0.003 | -0.003 | -0.321 | -0.068 | -0.044 | -0.318 | 0.420 | **0.505** |
| | H=336 | -0.445 | -0.296 | -0.296 | -0.791 | -0.029 | -0.198 | -0.025 | **0.440** | 0.016 |
| | H=720 | -0.637 | -0.244 | -0.244 | -0.178 | 0.070 | -0.023 | -0.345 | **0.452** | 0.442 |
| avg | H=96 | 0.008 | 0.020 | 0.056 | 0.318 | 0.003 | 0.069 | 0.319 | 0.031 | **0.470** |
| | H=192 | -0.046 | -0.027 | 0.006 | 0.212 | 0.023 | 0.079 | 0.301 | 0.104 | **0.453** |
| | H=336 | -0.201 | -0.066 | -0.084 | -0.004 | 0.066 | 0.045 | 0.294 | 0.114 | **0.411** |
| | H=720 | -0.238 | -0.090 | -0.054 | 0.073 | 0.040 | 0.059 | 0.271 | 0.251 | **0.432** |
| Num.Top-1 | | 0 | 3 | 2 | 8 | 4 | 0 | 6 | 5 | **28** |

2021)), and economics (Exchange (Lai et al., 2018)). Dataset statistics are detailed in Appendix A.1. To ensure reproducibility, we follow standard dataset splits: training and validation sets are used for model selection, while the test set is reserved for evaluating ground-truth fine-tuning performance.

**Pre-trained Models.** To ensure robust evaluation, we select eight state-of-the-art time series pre-trained models from diverse architectures and training paradigms: (1) Encoder-only: MOIRAI (Woo et al., 2024), UniTS (Gao et al., 2024), and Moment (Goswami et al., 2024); (2) Decoder-only: TimesFM (Das et al., 2024) and Timer (Liu et al., 2024); (3) Encoder-decoder: TTM (Ekambaram et al., 2024), ROSE (Wang et al., 2024), and Chronos (Ansari et al., 2024). We collect ground-truth fine-tuning results from the TSFM-Bench benchmark (Li et al., 2025) for common forecasting horizons of $\{96, 192, 336, 720\}$, with the full results provided in the Appendix A.9.

Table 2: Methods comparison of Pr(top-$k$) and average $\tau_\omega$ across horizons on 14 datasets.

|  | Pr(top1) | Pr(top2) | Pr(top3) | $\tau_\omega$ |
|---|---|---|---|---|
| RankME | 0.000 | 0.000 | 0.196 | -0.119 |
| LogME | 0.071 | 0.196 | 0.268 | -0.041 |
| Regression | 0.036 | 0.125 | 0.286 | -0.019 |
| Etran | 0.304 | 0.393 | 0.536 | 0.150 |
| DISCO | 0.232 | 0.375 | 0.536 | 0.033 |
| Model Spider | 0.304 | 0.482 | 0.571 | 0.296 |
| zero shot | 0.286 | 0.464 | 0.589 | 0.125 |
| SwiftTS | **0.339** | **0.500** | **0.607** | **0.442** |

Table 3: Ablation studies for model embedding: $\tau_\omega$ and average across horizons are listed below.

| $v_a$ | $v_t$ | $v_c$ | 96 | 192 | 336 | 720 | avg |
|---|---|---|---|---|---|---|---|
| ✓ |  |  | 0.341 | 0.283 | 0.331 | 0.401 | 0.339 |
|  | ✓ |  | 0.225 | 0.270 | 0.318 | 0.227 | 0.267 |
|  |  | ✓ | 0.365 | 0.401 | 0.317 | 0.397 | 0.370 |
| ✓ | ✓ |  | 0.361 | 0.383 | 0.315 | 0.417 | 0.369 |
|  | ✓ | ✓ | 0.427 | 0.430 | 0.328 | 0.391 | 0.394 |
| ✓ |  | ✓ | 0.380 | 0.422 | **0.437** | 0.403 | 0.411 |
| ✓ | ✓ | ✓ | **0.470** | **0.453** | 0.411 | **0.432** | **0.442** |

**Baselines and Metrics.** We compare various model selection methods under three paradigms: (1) Feature-analytic methods: RankME (Garrido et al., 2023), LogME (You et al., 2021), Regression (Gholami et al., 2023), Etran (Gholami et al., 2023), DISCO (Zhang et al., 2025). (2) Learning-based method: Model Spider (Zhang et al., 2023), AutoForecast (Abdallah et al., 2022). (3) Brute-force method: Zero-shot performance. Further details are provided in Appendix A.2. For evaluation, we use weighted Kendall's $\tau_\omega$ to measure the correlation between estimated scores $\{\hat{r}_k\}_{k=1}^K$ and fine-tuned results $\{r_k\}_{k=1}^K$, following previous work (Shao et al., 2022; Gholami et al., 2023; Li et al., 2023; Zhang et al., 2025). A larger $\tau_\omega$ indicates stronger alignment between estimated and true rankings. Details are in Appendix A.3.

**Implementation.** To assess generalization on unseen tasks and prevent data leakage, we adopt a strict splitting protocol: in each training run, we randomly select 3 out of the 14 benchmark datasets held out for testing, while the remaining 11 are split into training and validation sets using an 8:2 ratio. During training, a multiple-subset sampling strategy is applied to each dataset to construct the meta-dataset $\mathcal{D}_{\text{meta}}$. Once sampling is completed, the data encoder receives a fixed set of subsets across runs to ensure reproducibility. During evaluation, the same multiple-subset sampling strategy is used. In the rare case where different subsets of the same dataset yield inconsistent rankings, we adopt a voting-based ensemble to obtain the final model ranking. All experiments are conducted on an NVIDIA GeForce RTX 3090 GPU with batch size 16 for 80 epochs, using $G = 4$ experts. Optimization is performed with Adam ($\beta_1 = 0.9$, $\beta_2 = 0.999$). In the meta-learning process, the inner-loop and outer-loop learning rates are $\alpha = 0.001$ and $\gamma = 0.005$. The loss trade-off coefficient is $\lambda = 0.7$, with sensitivity analysis in Section A.8.

## 5.2 EXPERIMENTAL RESULTS

**Main Results.** Table 1 compares SwiftTS with various baselines across 14 datasets and four horizons of $\{96, 192, 336, 720\}$. The results demonstrate that SwiftTS outperforms the baselines in average $\tau_\omega$ across all horizons, highlighting its effectiveness in selecting high-performing pre-trained models. Feature-analytic methods often suffer from inconsistent features derived from pre-trained models with diverse architectures and paradigms. Model Spider alleviates this issue by learning a similarity function between datasets and models, but it overlooks sequential dependencies in time series and prior knowledge of the models. In contrast, SwiftTS employs a dual architecture consisting of a temporal-aware data encoder and a knowledge-infused encoder, which together enhance selection performance. Additionally, SwiftTS features a horizon-adaptive expert composition module, allowing it to handle multiple horizons simultaneously within a unified framework. By comparison, all baselines require recomputation or retraining for different horizons. This efficiency makes SwiftTS well-suited for real-world applications demanding flexible multi-horizon forecasting. Moreover, while other methods exhibit varying degrees of negative correlation in different datasets and horizons, SwiftTS adopts transferable cross-task learning to maintain predominantly positive correlations across 14 datasets and different horizons, showing its OOD robustness.

**Top-$k$ Performance.** We report the top-$k$ selection probability Pr(top-$k$) as used in (Zhang et al., 2025; Gholami et al., 2023). This metric evaluates the likelihood that the best-performing model appears within the top-$k$ of the estimated ranking. Results in Table 2 demonstrate that SwiftTS achieves the best top-$k$ performance, validating the model selection effectiveness of our method.

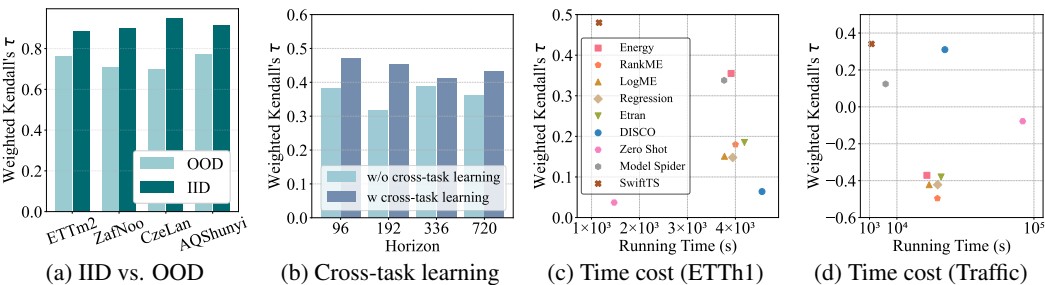

Figure 4: (a) Comparison of (a) average $\tau_\omega$ for IID vs. OOD settings across four datasets and horizons, and (b) ablation study of cross-task learning, and method comparison w.r.t running time (second) and average $\tau_\omega$ across horizons on (c) ETTh1 (small-scale) and (d) Traffic (large-scale).

## 5.3 FURTHER ANALYSIS

**Embedding ablation.** The model encoder integrates three embeddings to represent a neural network: meta-information embedding ($v_a$), topological embedding ($v_t$), and functional embeddings ($v_c$). To assess their individual contributions, we perform ablation studies by removing one or two embedding type at a time and measuring the resulting performance. As shown in Table 3, meta-information embedding $v_a$ and the functional embedding $v_c$ contribute the most to performance. The best results are obtained when all three embeddings are utilized together, demonstrating the complementary nature and necessity of each type of embedding.

**IID vs. OOD.** In Figure 4a, we compare the average $\tau_\omega$ performance of four forecasting horizons under IID and OOD settings across four datasets. Compared to OOD, the IID setting refers to evaluation on test data drawn from the same distribution as the training data. The results show that our method achieves further improvements under the IID setting, indicating that increasing the diversity of training data can enhance model performance.

**Cross-task learning ablation.** To validate the effectiveness of the proposed transferable cross-task learning, we conduct an ablation experiment as illustrated in Figure 4b. Specifically, we evaluate the model's average performance on 14 downstream datasets across four horizons. A clear advantage is observed when comparing the results from the w/o cross-task learning setting to those with the cross-task learning enabled. This consistent improvement across horizons demonstrates that our cross-task learning effectively enhances the model's predictive capability and robustness.

**Efficiency and Scalability.** We evaluate efficiency by analyzing runtime on small (ETTh1) and large (Traffic) datasets (Figure 4c, Figure 4d). Results show that model selection methods significantly reduce computational overhead compared to full fine-tuning. For example, on ETTh1, model selection methods typically require only 1,000 to 4,000 seconds, whereas fully fine-tuning each model takes approximately $4.97 \times 10^4$ seconds on the same GPU. This stark contrast highlights the critical importance of efficient model selection methods in practical applications. Moreover, the comparison across different dataset scales reveals that the runtime cost of SwiftTS remains relatively stable and is less sensitive to dataset size. In contrast, the time overhead of other model selection baselines increases dramatically as the dataset grows. The cost of fine-tuning on the Traffic dataset reaches up to $3.46 \times 10^6$ seconds, making it prohibitively expensive for large-scale applications. Overall, SwiftTS achieves both superior performance and minimal time overhead.

## 6 CONCLUSION

This paper tackles the challenge of pre-trained model selection from a model hub for time series forecasting. We propose SwiftTS, a learning-guided framework with a lightweight dual-encoder architecture that independently embeds time series and candidate models, computing patchwise compatibility scores for efficient selection. To further enhance adaptability, SwiftTS incorporates a horizon-adaptive expert composition module for multi-task forecasting and leverages transferable cross-task learning to improve generalization across datasets. Extensive experiments show that SwiftTS achieves SOTA with high efficiency and scalability for real-world deployment.

ACKNOWLEDGMENTS

This work was supported by the National Natural Science Foundation of China (62406112, 62306085, 62372179) and Shenzhen College Stability Support Plan (GXWD202311130151329002).

ETHICS STATEMENT

This work is conducted entirely on publicly available benchmark datasets, as detailed in the paper, and does not involve the release of any personal or sensitive information. No human subjects are involved in this research, ensuring that our work complies with ethical standards in research integrity.

REPRODUCIBILITY STATEMENT

The performance of SwiftTS and the datasets used in our work are real, and all experimental results can be reproduced, as detailed in the paper. Details of model architecture, training procedures, and evaluation protocols are provided in the main text and appendix. To further facilitate reproducibility, we release the code and datasets at https://github.com/decisionintelligence/SwiftTS.

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

# A APPENDIX

## A.1 DATASETS

We evaluate SwiftTS on 14 multivariate time-series datasets spanning six distinct domains, including electricity, energy, traffic, environment, nature, and economics. (1) The ETT datasets (Zhou et al., 2021) contain 7 variables collected from two different power transformers between July 2016 and July 2018. The dataset consists of four subsets: ETTh1 and ETTh2, recorded hourly, and ETTm1 and ETTm2, recorded at 15-minute intervals. (2) Electricity (Trindade, 2015) records hourly electricity consumption from 321 customers over three years, from July 2016 to July 2019. (3) Solar (Lai et al., 2018) captures solar power generation from 137 photovoltaic plants in 2006, sampled every 10 minutes. (4) Wind (Li et al., 2022) consists of historical wind measurements (e.g., speed, direction). (5) PEMS08 (Song et al., 2020) contains three months of aggregated statistics on traffic flow, speed, and occupancy rate. (6) Traffic (Wu et al., 2021) contains hourly road occupancy rates measured by 862 sensors across freeways in the San Francisco Bay Area from 2015 to 2016. (7) Weather (Wu et al., 2021) includes 21 meteorological variables (e.g., temperature, humidity, and barometric pressure), recorded every 10 minutes across Germany in 2020. (8) AQShunyi (Zhang et al., 2017) provides hourly temperature measurements exhibiting strong seasonal patterns. (9) ZafNoo (Poyatos et al., 2021) is collected from the Sapflux data project and includes sap flow measurements and environmental variables. (10) CzeLan (Poyatos et al., 2021) is from the Sapflux data project, including sap flow measurements and environmental variables. (11) Exchange (Lai et al., 2018) comprises daily exchange rates for eight countries over a multi-year period. Table 4 summarizes the key statistics of these 14 multivariate time series datasets.

Table 4: Statistics of datasets.

| Dataset | Domain | Frequency | Lengths | Dim | Split |
|---------|--------|-----------|---------|-----|-------|
| ETTh1 (Zhou et al., 2021) | Electricity | 1 hour | 14,400 | 7 | 6:2:2 |
| ETTh2 (Zhou et al., 2021) | Electricity | 1 hour | 14,400 | 7 | 6:2:2 |
| ETTm1 (Zhou et al., 2021) | Electricity | 15 mins | 57,600 | 7 | 6:2:2 |
| ETTm2 (Zhou et al., 2021) | Electricity | 15 mins | 57,600 | 7 | 6:2:2 |
| Electricity (Trindade, 2015) | Electricity | 1 hour | 26,304 | 321 | 7:1:2 |
| Solar (Lai et al., 2018) | Energy | 10 mins | 52,560 | 137 | 6:2:2 |
| Wind (Li et al., 2022) | Energy | 15 mins | 48,673 | 7 | 7:1:2 |
| PEMS08 (Song et al., 2020) | Traffic | 5 mins | 17,856 | 170 | 6:2:2 |
| Traffic (Wu et al., 2021) | Traffic | 1 hour | 17,544 | 862 | 7:1:2 |
| Weather (Wu et al., 2021) | Environment | 10 mins | 52,696 | 21 | 7:1:2 |
| AQShunyi (Zhang et al., 2017) | Environment | 1 hour | 35,064 | 11 | 6:2:2 |
| ZafNoo (Poyatos et al., 2021) | Nature | 30 mins | 19,225 | 11 | 7:1:2 |
| CzeLan (Poyatos et al., 2021) | Nature | 30 mins | 19,934 | 11 | 7:1:2 |
| Exchange (Lai et al., 2018) | Economic | 1 day | 7,588 | 8 | 7:1:2 |

## A.2 BASELINES

In our study, we compare various model selection methods for pre-trained time series models, which can be broadly categorized into three paradigms: **(1) Feature-analytic methods**: These methods rely on intrinsic properties or statistical characteristics of features extracted by the pre-trained models to estimate their transferability. **RankME** (Garrido et al., 2023) evaluates the rank of feature matrices extracted from the model's representations. **LogME** (You et al., 2021) computes the logarithm of the maximum label marginalized likelihood under a probabilistic model. **Regression** (Gholami et al., 2023) employs linear regression using Singular Value Decomposition (SVD) to approximate the mapping from model features to target labels. **Etran** (Gholami et al., 2023) combines both the energy score and the regression score into a unified metric. **DISCO** (Zhang et al., 2025) evaluates pre-trained models by analyzing the spectral distribution of their feature representations, enabling the assessment in both classification and regression tasks. **(2) Learning-based method**: **Model Spider** (Zhang et al., 2023) learns model representations and a similarity function through alignment with downstream task representations, facilitating model selection via the learned similarity.

**AutoForecast** (Abdallah et al., 2022) leverages the performance tensor and the meta-feature tensor to predict model performance under different hyperparameters. **(3) Brute-force method**: **Zero-shot** measures a model's ability to generalize to unseen tasks without any task-specific fine-tuning, offering valuable insights into its overall generalization capacity.

## A.3   METRICS

Kendall's $\tau$ measures the ordinal association between two rankings by evaluating the number of concordant and discordant pairs, which is defined as:

$$\tau = \frac{2}{K(K-1)} \sum_{1 \le i < j \le K} sgn(r^i - r^j) sgn(\hat{r}^i - \hat{r}^j) \tag{11}$$

where $sgn(x)$ is the sign function. However, in practical model selection scenarios, accurately identifying the top-performing models is often more critical than precisely ranking lower-performing ones. To reflect this priority, we employ the weighted version of Kendall's $\tau$, denoted as $\tau_\omega$. This variant adjusts the contribution of each pairwise comparison by assigning larger weights to higher-ranked models. A higher value of $\tau_\omega$ indicates stronger consistency between estimated and actual rankings, reflecting the reliability of the evaluation metric in guiding model selection.

## A.4   ALGORITHM

---

**Algorithm 1:** Pseudo-code of `SwiftTS`

---

1  **Training**:
2  **Input:** Meta-dataset $\mathcal{D}_{\text{meta}} = \{D^i, Z, H^i, r^i\}_{i=1}^N$; Total training epochs $E$; Inner-loop learning rate $\alpha$; Outer-loop learning rate $\gamma$ ;
3  Randomly initialize the weights $\theta$ of the whole model;
4  **for** $e \leftarrow 1$ **to** $E$ **do**
5  $\quad$ $\mathcal{T} \leftarrow$ Sample $n$ tasks from $\mathcal{D}_{\text{meta}}$;
6  $\quad$ **for** $j \leftarrow 1$ **to** $n$ **do**
7  $\quad\quad$ $support_j, query_j \leftarrow$ Obtain support set and query set from $\mathcal{T}_j$;
8  $\quad\quad$ $\hat{r} \leftarrow$ `ModelRanking`$(Z, support_j, H)$
9  $\quad\quad$ Evaluate $\nabla_\theta \mathcal{L}_{\text{supp}}(\mathcal{T}_j; \theta)$ using 6;
10 $\quad\quad$ Compute $\theta'_j \leftarrow \theta - \alpha \nabla_\theta \mathcal{L}_{\text{supp}}(\mathcal{T}_j; \theta)$;
11 $\quad$ Update $\theta \leftarrow \theta - \gamma \nabla_\theta \sum_{\mathcal{T}_j} \mathcal{L}_{\text{query}}(\mathcal{T}_j; \theta'_j)$;

12 **Inference**:
13 **Input:** An unseen downstream dateset $X$, model hub $Z$, forecasting horizon $H$;
14 $\hat{r} \leftarrow$ `ModelRanking`$(Z, X, H)$
15 **Return:** The predicted ranking scores $\hat{r}$ for $K$ candidate pre-trained models in the model hub.

16 **Function** `ModelRanking`$(Z, X, H)$:
17 $\quad$ $E_d, E_m \leftarrow E_D(X), E_M(Z)$;
18 $\quad$ $E_{\text{ca}} \leftarrow$ Compute patchwise CA using Eq. 4;
19 $\quad$ $w \leftarrow$ Compute the weights that router assigns to each expert using Eq. 7 with $H$;
20 $\quad$ **Return:** $\hat{r} \leftarrow$ Rank pre-trained models using Eq. 8;

---

## A.5   DETAILS OF META-INFORMATION OF PRE-TRAINED MODELS

The five types of meta-information of pre-trained models include: model architecture (category), model capacity (scalar), model complexity (scalar), model dimension (scalar), and pre-trained domain (category). Categorical features are converted into one-hot vectors. For the pre-trained domain, which may involve multiple labels, a multi-label one-hot vector is employed to represent each domain. Scalar features are normalized based on their minimum and maximum values across all

Table 5: Embedding computation runtime (in seconds) for each model.

| Embedding | TimesFM | UniTS | Moment | TTM | Moirai | Rose | Timer | Chronos |
|---|---|---|---|---|---|---|---|---|
| functional | 0.714 | 0.335 | 0.578 | 0.247 | 0.606 | 0.438 | 0.396 | 0.683 |
| topological | 1.092 | 0.471 | 2.328 | 0.317 | 1.084 | 0.532 | 0.490 | 1.720 |

models. The resulting normalized scalar features and one-hot encoded categorical features are then concatenated to construct the meta-information embedding. Since the meta-information embedding may have a semantic gap with the topological and functional embedding, we apply a linear projection to map it into a shared latent space. The projected embedding is subsequently integrated with the topological and functional embeddings, as described in Equation 3, to generate the final model representation used in our model selection framework.

### A.6    EFFICIENCY ANALYSIS OF TOPOLOGICAL AND FUNCTIONAL EMBEDDINGS.

The construction of both topological and functional embeddings is an offline, dataset-agnostic process. It only requires feeding each candidate model a fixed input with the correct dimensionality (e.g., a fixed set of Gaussian random noise vectors, rather than full downstream datasets). This incurs a total preprocessing cost of $\mathcal{O}(N \cdot T_{emb})$, where $N$ is the number of candidate models and $T_{emb}$ denotes the time required to compute the embeddings for a single model. We empirically evaluate the runtime (in seconds) required to obtain these embeddings for each model in Table 5.

In contrast, existing approaches require substantial forward-pass computation over the entire downstream dataset for every candidate model. This results in a total cost of $\mathcal{O}(N \cdot T_f(D))$. Here, $T_f(D) = \Omega(|D| \cdot C_{model})$, where $|D|$ is size of dataset and $C_{model}$ represents the per-sample computational cost. Thus, their runtime depends not only on model complexity but also heavily on the size of the downstream data. For example, on the same hardware, the forward-pass runtime of TimesFM (200M) is 654.98 seconds on the relatively small AQShunyi dataset, but increases to 5,426.12 seconds on the larger Solar dataset. Overall, because $T_{emb} \ll T_f(D)$ and the embedding computation is performed offline, the linear cost growth with respect to $N$ remains highly tractable. The per-model embedding overhead of SwiftTS is minimal and does not introduce a resource bottleneck as the candidate models increases. Moreover, since embedding computation is fully offline, it can be precomputed if still concerns about runtime.

### A.7    SCALABILITY OF THE MODEL HUB

To investigate how the number of candidate models affects the performance of SwiftTS, we augment the original model hub by adding six additional models: Chronos-mini and Chronos-small (the hub originally contained Chronos-base) (Ansari et al., 2024), Moirai-small and Moirai-large (originally Moirai-base) (Woo et al., 2024), as well as TimeMoE-base and TimeMoE-large(Shi et al., 2025). This expansion results in a total of 14 candidate models in the model hub. We then compare SwiftTS against existing baselines under these more challenging conditions. The performance, measured by the average weighted Kendall's $\tau_\omega$ across horizons on 14 target datasets, is presented in Table 6. The results show that increasing the size and heterogeneity of the model pool indeed makes the model selection task more challenging, leading to a certain degree of performance degradation for all methods. However, our framework exhibits notably lower sensitivity to this expansion and continues to outperform existing baselines.

### A.8    MORE EXPERIMENTAL RESULTS

**The perforamnce of Top1-selected model.** We report the actual forecasting performance of the top1-selected model for each selection method, with the average MSE across horizons as shown in Table 7. The "Best model" column denotes the performance of the ground-truth top1 model, representing the upper bound achievable by any selection method. The results demonstrate that SwiftTS not only selects pre-trained models effectively but also delivers superior actual forecasting performance across a wide range of datasets.

Table 6: Scalability analysis of the model hub with an expanded and more diverse set of 14 candidate models. The average weighted Kendall's $\tau_\omega$ across horizons on 14 datasets is listed below.

| | RankME | LogME | Regression | Etran | DISCO | Model spider | zero-shot | **SwiftTS** |
|---|---|---|---|---|---|---|---|---|
| ETTh1 | -0.207 | 0.103 | 0.074 | 0.207 | 0.106 | 0.236 | 0.274 | **0.402** |
| ETTh2 | -0.066 | -0.018 | -0.056 | 0.175 | 0.109 | 0.347 | 0.248 | **0.423** |
| ETTm1 | 0.050 | 0.131 | 0.111 | 0.217 | 0.065 | 0.149 | 0.290 | **0.415** |
| ETTm2 | -0.029 | 0.022 | 0.075 | 0.303 | 0.208 | 0.151 | 0.304 | **0.453** |
| Electricity | 0.174 | -0.019 | 0.132 | 0.100 | 0.043 | 0.089 | 0.188 | **0.383** |
| Traffic | 0.069 | 0.081 | 0.086 | 0.065 | 0.341 | 0.175 | 0.190 | **0.453** |
| Solar | 0.005 | 0.190 | 0.105 | 0.189 | 0.090 | **0.243** | 0.139 | 0.240 |
| Weather | 0.121 | 0.080 | 0.032 | **0.251** | 0.227 | 0.173 | 0.209 | 0.238 |
| Exchange | 0.032 | 0.019 | 0.151 | 0.129 | 0.272 | 0.095 | 0.024 | **0.286** |
| ZafNoo | -0.150 | -0.020 | 0.087 | -0.016 | 0.092 | 0.440 | 0.186 | **0.649** |
| CzeLan | -0.257 | -0.076 | -0.110 | 0.150 | 0.047 | 0.333 | 0.219 | **0.515** |
| AQShunyi | -0.077 | 0.004 | -0.012 | 0.021 | 0.269 | 0.455 | 0.438 | **0.575** |
| Wind | -0.242 | 0.176 | 0.189 | 0.228 | 0.039 | 0.181 | 0.056 | **0.310** |
| PEMS08 | 0.112 | 0.037 | 0.009 | 0.092 | 0.097 | 0.267 | 0.302 | **0.373** |

Table 7: Performance of the top1-selected model. The average MSE across horizons is listed below.

| | RankME | LogME | Regression | Etran | DISCO | Model spider | zero-shot | SwiftTS | Best model |
|---|---|---|---|---|---|---|---|---|---|
| ETTh1 | 0.404 | 0.404 | 0.403 | 0.421 | 0.425 | 0.403 | 0.413 | **0.393** | 0.391 |
| ETTh2 | 0.349 | 0.345 | 0.343 | 0.343 | 0.352 | 0.347 | 0.342 | **0.340** | 0.331 |
| ETTm1 | 0.735 | 0.345 | 0.345 | 0.346 | 0.436 | 0.383 | 0.345 | **0.341** | 0.340 |
| ETTm2 | 0.304 | 0.258 | 0.258 | 0.251 | 0.276 | 0.269 | 0.253 | **0.251** | 0.246 |
| Electricity | 0.250 | 0.212 | 0.212 | 0.169 | 0.195 | 0.196 | 0.250 | **0.163** | 0.155 |
| Traffic | 0.539 | 0.668 | 0.668 | **0.390** | 0.393 | 0.555 | 0.555 | **0.390** | 0.380 |
| Solar | 0.252 | **0.186** | **0.186** | 0.552 | 0.905 | 0.252 | 0.358 | 0.252 | 0.180 |
| Weather | 0.269 | 0.250 | 0.245 | **0.236** | 0.243 | 0.243 | 0.255 | **0.236** | 0.217 |
| Exchange | 0.407 | 0.477 | 0.484 | 0.490 | **0.388** | 0.407 | 0.419 | 0.406 | 0.362 |
| ZafNoo | 0.608 | 0.561 | 0.589 | 0.522 | 0.533 | 0.542 | 0.513 | **0.511** | 0.502 |
| CzeLan | 0.755 | 0.695 | 0.755 | 0.232 | 0.326 | 0.606 | 0.217 | **0.206** | 0.206 |
| AQShunyi | 0.756 | 0.736 | 0.736 | 0.738 | 0.698 | 0.756 | 0.692 | **0.681** | 0.681 |
| Wind | 1.194 | 1.118 | 1.163 | **1.101** | 1.372 | 1.194 | 1.243 | 1.138 | 1.101 |
| PEMS08 | 0.708 | 0.687 | 0.687 | 0.918 | 0.382 | 0.708 | **0.255** | 0.319 | 0.241 |
| avg | 0.538 | 0.496 | 0.505 | 0.479 | 0.495 | 0.490 | 0.436 | **0.402** | 0.381 |

**Sensitivity Analysis.** We examine the impact of $\lambda \in [0.0, 0.3, 0.5, 0.7, 0.9, 1.0]$ and report average results across all datasets. As shown in Figure 5a, the performance remains relatively stable due to the complementary roles of the two loss components: $\mathcal{L}_{\text{rank}}$ constrains the relative ranking among pre-trained models, while $\mathcal{L}_{\text{pred}}$ enforces an absolute constraint on the gap between predicted performance and actual fine-tuning results. The experimental results highlight the indispensable role of the prediction loss and suggest that the best performance is achieved when $\lambda$ is set to $0.7$.

**Choices of Expert Numbers.** We study the impact of expert number $G \in [2, 4, 6, 8, 10]$ ( Figure 5b). A small $G$ fails to adequately capture the differences across horizons, limiting its ability to perform effective multi-task forecasting. Conversely, a large $G$ increases computational cost and model complexity, which may lead to overfitting and degraded performance on downstream tasks. Overall, $G = 4$ provides the best trade-off between accuracy and efficiency, making it ideal for practical use.

**Visualization.** We visualize the patchwise cross-attention weights in Figure 6, which reveal several key observations: (1) Figure 6a and Figure 6b compare the data embeddings $E_d$ obtained from different ETTh2 subsets. We observe that the weight distributions across different patches for the same model (same row) are quite similar. Furthermore, TTM and ROSE, the two best-performing models on ETTh2, also exhibit similar patchwise weight distributions. This suggests that models with comparable performance tend to share similar patchwise attention patterns within the same dataset. (2) When comparing $E_d$ from two similar datasets, ETTh2 (Figure 6b) and ETTh1 (Figure 6c), we find that the patchwise weight distributions for the same model remain relatively consistent. This indicates that similar attention patterns are exhibited not only across different subsets of the same dataset, but also between similar datasets. (3) In contrast, when comparing two more distinct datasets, ETTh1 (Figure 6c) and PEMS08 (Figure 6d), significant differences emerge in the patchwise weight distributions of the same model. Moreover, among the top-performing models on PEMS08, such as Moirai, TimesFM, Moment, similar patchwise distributions are evident. The best model, Moirai, places the highest attention on the 8-th patch, whereas two weaker models, TTM and

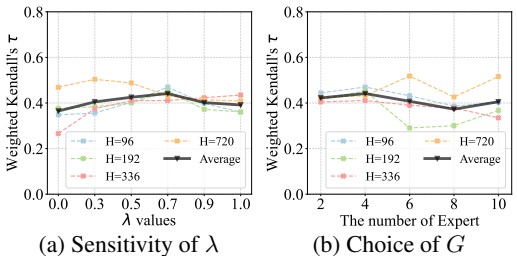

Figure 5: (a) Sensitivity analysis of the loss coefficient $\lambda$, (b) choice of the number of experts $G$.

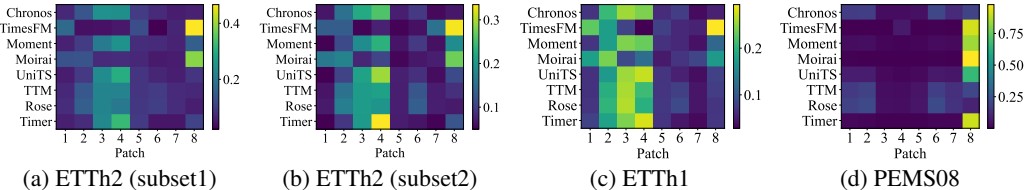

Figure 6: Visualization of patchwise cross-attention weights.

ROSE, show considerably lower weights for that patch. This indicates that models with different performance levels tend to focus on different patches.

## A.9 GROUND-TRUTH FINE-TUNING RESULTS

We obtain the ground-truth forecasting results of the pretrained models after fine-tuning from the TSFM-Bench benchmark to ensure fairness and reproducibility. For completeness, we further report the results in Table 8.

Table 8: The ground-truth fine-tuning results of the pre-trained models in the model hub. The results are MSE of each prediction length.

| Dataset | H | Chronos | TimesFM | Moment | UniTS | Moirai | TTM | Rose | Timer |
|---|---|---|---|---|---|---|---|---|---|
| ETTh1 | 96 | 0.388 | 0.373 | 0.383 | 0.399 | 0.394 | 0.361 | 0.354 | 0.416 |
| | 192 | 0.440 | 0.418 | 0.415 | 0.441 | 0.430 | 0.393 | 0.389 | 0.557 |
| | 336 | 0.477 | 0.457 | 0.425 | 0.503 | 0.450 | 0.411 | 0.406 | 0.502 |
| | 720 | 0.475 | 0.458 | 0.447 | 0.468 | 0.457 | 0.426 | 0.413 | 0.525 |
| ETTh2 | 96 | 0.292 | 0.288 | 0.287 | 0.311 | 0.285 | 0.270 | 0.265 | 0.305 |
| | 192 | 0.362 | 0.371 | 0.350 | 0.470 | 0.352 | 0.338 | 0.328 | 0.394 |
| | 336 | 0.404 | 0.418 | 0.367 | 0.429 | 0.384 | 0.367 | 0.353 | 0.414 |
| | 720 | 0.412 | 0.441 | 0.404 | 0.424 | 0.419 | 0.384 | 0.376 | 0.521 |
| ETTm1 | 96 | 0.339 | 0.313 | 0.287 | 0.321 | 0.464 | 0.285 | 0.275 | 0.344 |
| | 192 | 0.392 | 0.353 | 0.326 | 0.373 | 0.488 | 0.325 | 0.324 | 0.447 |
| | 336 | 0.440 | 1.177 | 0.353 | 0.388 | 0.520 | 0.357 | 0.354 | 0.457 |
| | 720 | 0.530 | 1.095 | 0.408 | 0.452 | 0.598 | 0.413 | 0.411 | 1.444 |
| ETTm2 | 96 | 0.181 | 0.172 | 0.170 | 0.198 | 0.224 | 0.165 | 0.157 | 0.188 |
| | 192 | 0.253 | 0.234 | 0.230 | 0.252 | 0.308 | 0.225 | 0.213 | 0.281 |
| | 336 | 0.318 | 0.357 | 0.283 | 0.334 | 0.369 | 0.275 | 0.266 | 0.328 |
| | 720 | 0.417 | 0.454 | 0.375 | 0.468 | 0.460 | 0.367 | 0.347 | 0.493 |
| Electricity | 96 | 0.133 | 0.142 | 0.148 | 0.133 | 0.170 | 0.132 | 0.125 | 0.136 |
| | 192 | 0.152 | 0.163 | 0.165 | 0.153 | 0.186 | 0.149 | 0.142 | 0.169 |
| | 336 | 0.171 | 0.332 | 0.182 | 0.175 | 0.205 | 0.270 | 0.162 | 0.196 |
| | 720 | 0.201 | 0.364 | 0.223 | 0.204 | 0.247 | 0.297 | 0.191 | 0.364 |
| Traffic | 96 | 0.385 | 0.419 | 0.383 | 0.377 | 0.358 | 0.379 | 0.354 | 0.362 |
| | 192 | 0.411 | 0.450 | 0.397 | 0.387 | 0.372 | 0.396 | 0.377 | 0.396 |
| | 336 | 0.521 | 0.939 | 0.407 | 0.395 | 0.380 | 0.945 | 0.396 | 0.427 |
| | 720 | 0.623 | 0.957 | 0.443 | 0.436 | 0.412 | 0.952 | 0.434 | 0.970 |
| Solar | 96 | 0.430 | 0.174 | 0.172 | 0.163 | 0.877 | 0.174 | 0.170 | 0.183 |
| | 192 | 0.396 | 0.198 | 0.187 | 0.176 | 0.928 | 0.181 | 0.204 | 0.225 |
| | 336 | 0.409 | 1.530 | 0.196 | 0.184 | 0.956 | 0.189 | 1.616 | 0.244 |
| | 720 | 0.453 | 1.322 | 0.206 | 0.196 | 1.016 | 0.200 | 0.215 | 0.355 |
| Weather | 96 | 0.183 | 0.161 | 0.152 | 0.147 | 0.206 | 0.149 | 0.145 | 0.164 |
| | 192 | 0.227 | 0.207 | 0.196 | 0.191 | 0.278 | 0.199 | 0.183 | 0.243 |
| | 336 | 0.286 | 0.311 | 0.245 | 0.243 | 0.335 | 0.256 | 0.232 | 0.321 |
| | 720 | 0.368 | 0.370 | 0.316 | 0.317 | 0.413 | 0.340 | 0.309 | 0.349 |
| Exchange | 96 | 0.093 | 0.086 | 0.085 | 0.444 | 0.096 | 0.113 | 0.086 | 0.104 |
| | 192 | 0.199 | 0.193 | 0.178 | 0.507 | 0.200 | 0.223 | 0.178 | 0.221 |
| | 336 | 0.370 | 0.354 | 0.333 | 0.489 | 0.381 | 0.439 | 0.341 | 0.382 |
| | 720 | 0.856 | 0.988 | 0.851 | 0.997 | 1.133 | 1.185 | 0.947 | 0.965 |
| ZafNoo | 96 | 0.463 | 0.457 | 0.430 | 0.444 | 0.439 | 0.426 | 0.431 | 0.470 |
| | 192 | 0.524 | 0.576 | 0.486 | 0.507 | 0.501 | 0.479 | 0.487 | 0.548 |
| | 336 | 0.575 | 0.650 | 0.530 | 0.563 | 0.551 | 0.523 | 0.538 | 0.588 |
| | 720 | 0.684 | 0.748 | 0.585 | 0.602 | 0.616 | 0.583 | 0.578 | 0.637 |
| CzeLan | 96 | 0.505 | 0.198 | 0.171 | 0.196 | 0.611 | 0.162 | 0.164 | 0.224 |
| | 192 | 0.565 | 0.244 | 0.201 | 0.226 | 0.623 | 0.192 | 0.198 | 1.198 |
| | 336 | 0.669 | 1.232 | 0.225 | 0.250 | 0.654 | 0.217 | 0.221 | 0.750 |
| | 720 | 0.838 | 1.214 | 0.264 | 0.323 | 0.702 | 0.253 | 0.253 | 0.848 |
| AQShunyi | 96 | 0.728 | 0.662 | 0.660 | 0.739 | 0.621 | 0.640 | 0.632 | 0.814 |
| | 192 | 0.802 | 0.746 | 0.707 | 0.784 | 0.665 | 0.683 | 0.677 | 0.882 |
| | 336 | 0.843 | 0.795 | 0.727 | 0.829 | 0.697 | 0.706 | 0.706 | 0.890 |
| | 720 | 0.897 | 0.820 | 0.782 | 0.857 | 0.740 | 0.763 | 0.770 | 0.953 |
| Wind | 96 | 1.177 | 0.913 | 0.915 | 0.949 | 0.957 | 0.889 | 0.904 | 1.087 |
| | 192 | 1.391 | 1.098 | 1.101 | 1.151 | 1.164 | 1.056 | 1.086 | 1.341 |
| | 336 | 1.540 | 1.326 | 1.231 | 1.329 | 1.333 | 1.189 | 1.238 | 1.514 |
| | 720 | 1.685 | 1.437 | 1.303 | 1.545 | 1.466 | 1.271 | 1.330 | 1.751 |
| PEMS08 | 96 | 0.804 | 0.167 | 0.261 | 0.519 | 0.144 | 0.177 | 0.199 | 0.194 |
| | 192 | 1.264 | 0.267 | 0.335 | 0.654 | 0.211 | 0.268 | 0.391 | 0.359 |
| | 336 | 1.317 | 1.285 | 0.365 | 0.599 | 0.276 | 1.206 | 1.441 | 0.385 |
| | 720 | 1.521 | 1.111 | 0.381 | 0.660 | 0.333 | 1.097 | 1.351 | 2.235 |

