# OpenReview forum: "SwiftTS: A Swift Selection Framework for Time Series Pre-trained Models via Multi-task Meta-Learning"
_ICLR.cc/2026/Conference — ICLR 2026 Poster_

### Official Review · Reviewer_SnbV · 2025-10-26

**Soundness:** 2
**Presentation:** 3
**Contribution:** 2
**Rating:** 2
**Confidence:** 3

**Summary:**

This paper introduces SwiftTS, a learning-guided framework for rapid selection of time series pre-trained models using a multi-task meta-learning strategy. The approach employs a dual-encoder (temporal-aware data encoder and knowledge-infused model encoder) to encapsulate both dataset and model features, leveraging patch-wise cross-attention to compute compatibility scores for model selection across various horizons. To improve generalization across datasets and forecasting horizons, the method incorporates horizon-adaptive expert composition as well as a cross-task meta-learning protocol. SwiftTS is evaluated on 14 real-world time series datasets and 8 pre-trained models, demonstrating strong results against a wide range of feature-analytic and learning-based baselines.

**Strengths:**

1.	The idea of selecting more suitable pretrained models for different downstream tasks and datasets is both interesting and valuable, addressing a practically important yet underexplored challenge in time-series foundation modeling.

2.	The dual-encoder architecture is well-motivated for heterogeneous time series model pools and directly addresses the issue of costly, inconsistent feature extraction in prior work. The use of a patch-wise attention mechanism reflects a careful design choice that captures local temporal structures relevant for model-dataset compatibility.

3.	SwiftTS introduces a horizon-adaptive expert composition module that flexibly and effectively addresses horizon-specific variability, as showcased in both the framework diagram and associated experimental tables.

**Weaknesses:**

1.	The paper emphasizes that SwiftTS avoids costly forward passes through all candidate models. Yet, the functional embedding module still requires each candidate model to be evaluated (albeit offline) on synthetic inputs such as Gaussian noise. This operation remains linearly proportional to the number of candidate models and does not scale well to continuously evolving model pools. The efficiency claim is therefore only partially valid and should be quantified more carefully.
2.	Does the sampling strategy used in the Temporal-Aware Data Encoder introduce randomness that leads to different results across runs? If so, how is this variance controlled or mitigated during training and evaluation?
3.	What exactly does the topological structure represent? Specifically, what are the semantic meanings of the nodes and edges in this context?
4.	The results only report the alignment metric between the estimated and true rankings, but not the actual forecasting performance of different selection methods. Without these results, it is difficult to assess the real effectiveness of the proposed selection strategy.
5.	What are the forecasting results of the pretrained models on these datasets without applying the selection process? Reporting these would provide a clear baseline for evaluating the benefit of the proposed selection mechanism.
6.	There exist many other meta-learning-based methods for time-series forecasting, such as AutoForecast [1], which should be included for comparison in the experimental section.
7.	Could the authors clarify the rationale and empirical justification for using Gaussian noise to generate the functional embeddings of candidate models? Would using real or synthetic time-series inputs affect the informativeness or stability of these embeddings?
8.	How sensitive are the framework’s predictions to the number of candidate models in the hub? Does performance degrade with larger, more heterogeneous model pools, and what are the observed scaling behaviors?
9.	Could the authors provide more clarity about the construction of meta-training tasks, especially the sampling policy for forecasting horizons and dataset divisions?
10.	Could the authors clarify how they handle cases where meta-training tasks share overlapping datasets or closely related forecasting horizons, which may cause distribution leakage or task redundancy?

[1] Abdallah M, Rossi R, Mahadik K, et al. Autoforecast: Automatic time-series forecasting model selection[C]//Proceedings of the 31st ACM International Conference on Information & Knowledge Management. 2022: 5-14.

**Questions:**

See weaknesses.

---

> ### Author Response · Authors · 2025-11-21
>
> Thank you for your suggestion. We try to address your concerns in the following ways:
>
> Weaknesses:
> 1. The construction of both topological and functional embeddings is an offline, dataset-agnostic process. It only requires feeding each candidate model a fixed input with the correct dimensionality (e.g., a fixed set of Gaussian random noise vectors, rather than full downstream datasets). This results in a total preprocessing cost of $\mathcal{O}(N \cdot T_{emb})$, where $N$ is the number of candidate models and $T_{emb}$ denotes the time required to compute the embeddings for a single model. We empirically evaluate the runtime (in seconds) required to obtain these embeddings for each model:
>
>     | Embedding    | TimesFM  |  UniTS   | Moment | TinyTimeMixer | Moirai |  Rose | Timer |Chronos  |
>     |------------|----------|----------|--------|-------|----------|----------|----------|----------|
>     | functional | 0.714     | 0.335   | 0.578  | 0.247  | 0.606   | 0.438     | 0.396      | 0.683      |
>     | topological| 1.092     | 0.471   | 2.328  | 0.317  | 1.084   | 0.532     | 0.490      | 1.720      |
>
>     In contrast, existing approaches require substantial forward-pass computation over the entire downstream dataset for every candidate model. This results in a total cost of $\mathcal{O}(N \cdot T_{f}(D))$. Here, $T_{f}(D)=\Omega(|D|\cdot C_{model})$, where $|D|$ is size of dataset and $C_{model}$ represents​ the per-sample computational cost. Thus, their runtime depends not only on model complexity but also heavily on the size of the downstream data. For example, on the same hardware, the forward-pass runtime of TimesFM (200M) is 654.98 seconds on the relatively small AQShunyi dataset, but increases to 5,426.12 seconds on the larger Solar dataset.
>
>     Overall, because $T_{emb} \ll T_{f}(D)$, the linear growth in cost with respect to $N$ remains highly tractable. The per-model embedding overhead of SwiftTS is negligible compared to the forward-pass cost of existing methods and does not introduce a practical bottleneck as the candidate models grow. Moreover, since embedding computation is fully offline, it can be precomputed if still concerns about runtime with the continuously evolving model pools.
>
>     In the revised paper, we provide the efficiency analysis of the topological and functional embeddings in Appendix A.6.
>
> 2. The multiple-subset sampling strategy for each dataset is applied during the construction of the meta-dataset $\mathcal{D_\text{meta}}$. Once the sampling is completed, the data encoder receives a fixed set of subsets across runs.
> During training, although this sampling strategy introduces a degree of randomness, the variance is effectively mitigated through mean pooling within each subset.
> This operation yields a compact data embedding that captures the shared temporal patterns within each subset while remaining robust to sample-level variability. The visualization of patch-wise cross-attention weights in Figures 6(a) and 6(b) further confirms this robustness: even though the sampled subsets drawn from the same original dataset differ, the weight distributions across different patches for the same model (within the same row) remain highly similar.
>
>     Moreover, by sampling multiple subsets from each original dataset, the resulting subsets collectively provide a more comprehensive representation of the overall dataset. Since model selection is conducted based on the aggregated compatibility scores across these diverse subsets rather than a single random subset, the process more faithfully reflects the dataset-level compatibility and effectively balances intra-dataset variance.
>
>     With respect to the final forecasting model ranking, different subsets may introduce slight variations in absolute performance predictions. However, these differences are minimal and do not affect the relative ranking of candidate models, which is the primary objective of our model selection task.
>
>     During evaluation, the same subset sampling strategy is applied consistently to all datasets. In the rare case where different subsets of the same dataset yield inconsistent rankings, we adopt a voting-based ensemble to obtain the final model ranking.
>
>     The control of multiple-subset sampling variance is further explained in the revised paper (lines 190-198 and 449-451).

---

> ### Author Response · Authors · 2025-11-21
>
> Weaknesses:
>
> 3. To clarify what the topological structure represents, we provide a detailed explanation of our DAG construction process. By leveraging the chain rule and gradient propagation maps, we trace how data flows through the network, thereby identifying the operations applied to the data and the directed paths along which the data travels.
> In this DAG, each node corresponds to a computational operation performed within the network (e.g., normalization, activation). The edges correspond to the directed paths along which the data are propagated, reflecting the computational dependencies and information flow within the model.
>
>     In the revised paper (lines 240–245), we provide a clearer explanation of the semantic meanings of the nodes and edges in the topological structure.
>
> 4. We report the actual forecasting performance of the top1-selected model for each selection method, with the average MSE across horizons as shown below. The "Best model" column denotes the performance of the ground-truth top1 model, representing the upper bound achievable by any selection method.
>
>     |          | RankME  | LogME   | Regression | Etran   | DISCO   | Model spider | zero-shot | SwiftTS    | Best model |
>     |----------|---------|---------|------------|---------|---------|--------------|-----------|---------| ---------|
>     | ETTh1    | 0.404   | 0.404   | $\underline{0.403}$| 0.421   | 0.425   | $\underline{0.403}$        | 0.413     | **0.393** | 0.391|
>     | ETTh2    | 0.349   | 0.345   | 0.343      | 0.343   | 0.352   | 0.347        | $\underline{0.342}$     | **0.340** | 0.331 |
>     | ETTm1    | 0.735   | $\underline{0.345}$   | $\underline{0.345}$      | 0.346   | 0.436   | 0.383  | $\underline{0.345}$     | **0.341** | 0.340|
>     | ETTm2    | 0.304   | 0.258   | 0.258      | $\underline{0.251}$   | 0.276   | 0.269        | 0.253     | **0.251** | 0.246 |
>     | Electricity | 0.250 | 0.212   | 0.212     | $\underline{0.169}$   | 0.195   | 0.196       | 0.250     | **0.163** | 0.155 |
>     | Traffic  | 0.539   | 0.668   | 0.668      | **0.390** | 0.393   | 0.555      | 0.555     | **0.390** | 0.380 |
>     | Solar    | 0.252   | **0.186** | **0.186**| 0.552   | 0.905   | $\underline{0.252}$      | 0.358     | $\underline{0.252}$ | 0.180 |
>     | Weather  | 0.269   | 0.250   | 0.245      | **0.236** | $\underline{0.243}$ | $\underline{0.243}$      | 0.255     | **0.236** | 0.217 |
>     | Exchange | 0.407   | 0.477   | 0.484      | 0.490   | **0.388** | 0.407      | 0.419     | $\underline{0.406}$   | 0.362 |
>     | ZafNoo   | 0.608   | 0.561   | 0.589      | 0.522   | 0.533   | 0.542        | $\underline{0.513}$     | **0.511** | 0.502 |
>     | CzeLan   | 0.755   | 0.695   | 0.755      | 0.232   | 0.326   | 0.606        | $\underline{0.217}$     | **0.206** | 0.206 |
>     | AQShunyi | 0.756   | 0.736   | 0.736      | 0.738   | 0.698   | 0.756        | $\underline{0.692}$     | **0.681** | 0.681
>     | Wind     | 1.194   | $\underline{1.118}$   | 1.163      | **1.101** | 1.372   | 1.194      | 1.243     | 1.138   | 1.101 |
>     | PEMS08   | 0.708   | 0.687   | 0.687      | 0.918   | 0.382   | 0.708        | **0.255** | $\underline{0.319}$   | 0.241 |
>     | avg   | 0.538   | 0.496   | 0.505      | 0.479   | 0.495   | 0.490        | $\underline{0.436}$  | **0.402**   | 0.381 |
>
>     The results demonstrate that SwiftTS not only selects pre-trained models effectively but also delivers superior actual forecasting performance across a wide range of datasets.
>
>     We incorporate the performance of the Top-1 selected model in Appendix A.8 of the revised paper.
>
> 5. As stated in our experimental design, we obtain the ground-truth forecasting results of the pretrained models after fine-tuning on these datasets from the TSFM-Bench benchmark to ensure fairness and reproducibility. These ground-truth fine-tuning results are already included in our released code. For completeness, we further report them in the Appendix A.9 of the revised paper.

---

> ### Author Response · Authors · 2025-11-21
>
> Weaknesses:
>
> 6. The meta-learner in AutoForecast leverages the performance tensor and the meta-feature tensor to predict model performance under different hyperparameter configurations. Our SwiftTS adopts a temporal-aware data encoder and a knowledge-infused model encoder to compute fine-grained compatibility of dataset-model pairs. We further introduce a multi-task meta-learning strategy with a horizon-adaptive expert composition to enhance generalization across datasets and forecasting horizons. The resulting comparisons are shown below and clearly demonstrate the effectiveness of our method:
>
>     |           | ETTh1  | ETTh2  | ETTm1   | ETTm2   | Electricity | Traffic | Solar   |
>     |-----------|--------|--------|---------|---------|-------------|---------|---------|
>     | Autoforecast | 0.259  | 0.231  | -0.060  | -0.217  | -0.077      | 0.155   | -0.142  |
>     | SwiftTS     | **0.480**  | **0.334**  | **0.481**   | **0.761**   | **0.379**       | **0.282**  | **0.209**   |
>
>     |           |  Weather | Exchange | ZafNoo | CzeLan | AQShunyi | Wind   | PEMS08 |
>     |-----------|--------|-----------|--------|--------|----------|--------|--------|
>     | Autoforecast| -0.188  |  -0.572   | 0.460  | 0.541  | 0.402    | 0.146  | -0.057 |
>     | SwiftTS     | **0.341**   |  **0.088**    | **0.710**   | **0.697**  | **0.774**    | **0.304**  | **0.341**  |
>
>     The complete results of the AutoForecast have been updated in the revised paper in Table 1.
>
> 7. The choice of Gaussian inputs aims to obtain a generic and data-agnostic embedding of the model’s functional characteristics. Although using real or synthetic time-series inputs can reflect how a model responds to the real-world temporal structures, it causes the resulting functional embedding to inadvertently inherit biases or domain-specific priors from the chosen probing data. For example, a model would yield a more favorable embedding than others simply because the probing inputs are similar to its pre-trained data, which is unfair to other models. Moreover, in practice, the downstream tasks and their associated temporal structures are unknown when the SwiftTS is trained. Relying on any specific type of real-world probing data could thus introduce unintended preferences into the model selection process. In contrast, random Gaussian noise serves as a neutral stimulus, allowing us to observe the model’s intrinsic input-output behavior without imposing external assumptions.
>
>     We clarify the design of the probing models with Gaussian noise in the revised paper (lines 253-258).

---

> ### Author Response · Authors · 2025-11-21
>
> Weaknesses:
>
> 8. To investigate how the number of candidate models affects the performance of SwiftTS, we augment the original model hub by adding six additional models: Chronos-mini and Chronos-small (the hub originally contained Chronos-base), Moirai-small and Moirai-large (originally Moirai-base), as well as TimeMoE-base and TimeMoE-large. We then compare our SwiftTS with existing approaches. The average weighted Kendall’s $\tau_\omega$ across horizons on 14 target datasets is reported below.
>
>     |          | RankME  | LogME   | Regression | Etran   | DISCO   | Model spider | zero-shot | SwiftTS |
>     |----------|---------|---------|------------|---------|---------|--------------|-----------|---------|
>     | ETTh1    | -0.207  | 0.103  | 0.074      | 0.207   | 0.106   | 0.236     | $\underline{0.274}$  | **0.402** |
>     | ETTh2    | -0.066  | -0.018  | -0.056     | 0.175   | 0.109   | $\underline{0.347}$ | 0.248     | **0.423** |
>     | ETTm1    | 0.050   | 0.131   | 0.111      | 0.217   | 0.065   | 0.149     | $\underline{0.290}$     | **0.415** |
>     | ETTm2    | -0.029  | 0.022   | 0.075      | 0.303   | 0.208   | 0.151     | $\underline{0.304}$     | **0.453** |
>     | Electricity | 0.174 | -0.019  | 0.132      | 0.100   | 0.043   | 0.089    | $\underline{0.188}$     | **0.383** |
>     | Traffic  | 0.069   | 0.081   | 0.086      | 0.065   | $\underline{0.341}$   | 0.175     | 0.190     | **0.453** |
>     | Solar    | 0.005   | 0.190   | 0.105      | 0.189   | 0.090   | **0.243** | 0.139     |$\underline{0.240}$   |
>     | Weather  | 0.121   | 0.080   | 0.032      | **0.251** | 0.227   | 0.173   | 0.209     | $\underline{0.238}$   |
>     | Exchange | 0.032   | 0.019   | 0.151      | 0.129   | $\underline{0.272}$   | 0.095     | 0.024     | **0.286** |
>     | ZafNoo   | -0.150  | -0.020  | 0.087      | -0.016  | 0.092   | $\underline{0.440}$     | 0.186     | **0.649** |
>     | CzeLan   | -0.257  | -0.076  | -0.110     | 0.150   | 0.047   | $\underline{0.333}$     | 0.219     | **0.515** |
>     | AQShunyi | -0.077  | 0.004   | -0.012     | 0.021   | 0.269   | $\underline{0.455}$ | 0.438     | **0.575**   |
>     | Wind     | -0.242  | 0.176   | 0.189      | $\underline{0.228}$ | 0.039   | 0.181   | 0.056     | **0.310** |
>     | PEMS08   | 0.112   | 0.037   | 0.009      | 0.092   | 0.097   | 0.267     | $\underline{0.302}$ | **0.373**   |
>
>     The results show that increasing the size and heterogeneity of the model pool indeed makes the model selection task more challenging, leading to a certain degree of performance degradation for all methods. However, our framework exhibits notably lower sensitivity to this expansion and continues to outperform existing baselines.
>
>     We incorporate the scalability analysis of the model hub in Appendix A.7 of the revised paper.

---

> ### Author Response · Authors · 2025-11-21
>
> Weaknesses:
>
> 9. We construct meta-training tasks by sampling from the meta-dataset $\mathcal{D_\text{meta}}= \lbrace D^{i}, Z, H^{i}, \boldsymbol{r}^{i} \rbrace_{i=1}^N$, where each task consists of a support set and a query set. To ensure that the resulting tasks are diverse and generalizable, we employ two sampling strategies: cross-dataset sampling and cross-horizon sampling.
>  - For cross-dataset sampling, we emphasize that the support and query sets are drawn from different datasets $D$ to encourage domain generalization. Specifically, the support set is constructed by first randomly selecting several datasets and then obtaining corresponding multiple subsets from each selected dataset to form inputs for the data encoder. The query set is generated in a similar manner but exclusively from a disjoint set of datasets.
>  - For cross-horizon sampling, we emphasize that the support and query sets are drawn from different forecasting horizons $H$ to improve adaptability to varying horizons. For example, the support set might include samples with horizons {336, 720}, whereas the query set is sampled from different horizons, such as {96, 192}.
>
>     In practice, downstream datasets and target forecasting horizons often exhibit greater diversity. To mimic more realistic and challenging conditions, we combine the above two strategies. For instance, in a single meta-training task, the support set may be sampled from three datasets and two forecasting horizons, while the query set is then randomly sampled from a disjoint collection of these datasets and horizons.
>
>     We provide these details of the construction of meta-training tasks in the revised paper (lines 321-331).
>
> 10. As mentioned above, cross-dataset sampling ensures that the support and query sets are drawn from disjoint datasets. In addition, we obtain multiple subsets from each original dataset to construct the inputs for the data encoder. This guarantees that, even when subsets originate from the same dataset, their resulting data embeddings are not identical. Such intra-dataset subset sampling increases task diversity and reduces redundancy across tasks. For forecasting horizons, we adopt four commonly used horizons in time series forecasting. Our cross-horizon sampling strategy ensures that the horizon distributions of the support and query sets differ. Finally, during evaluation, we strictly hold out all downstream evaluation datasets from the training process, preventing data leakage between training and evaluation and enabling a more reliable assessment of generalization.
>     We clarify the concerns of distribution leakage and task redundancy of meta-training tasks in the revised paper (lines 332-339).

---

> ### Author Response · Authors · 2025-11-28
> **Looking forward to your feedback**
>
> Dear Reviewer SnbV,
>
> Thank you sincerely for your time and thoughtful feedback on our paper. With the discussion period coming to a close, we would like to confirm whether our responses have adequately addressed your main concerns.
>
> We have carefully considered all your comments and devoted substantial effort to revising the paper accordingly. Should any questions or suggestions remain, we would deeply appreciate the opportunity to address them and provide any further information needed.
>
> If you feel that our responses have resolved your concerns, we would also be thankful if you could take this into consideration when updating your evaluation.
>
> Thank you once again for your constructive feedback and support.
>
> Best regards,
>
> The Authors

---

### Official Review · Reviewer_ERT9 · 2025-10-27

**Soundness:** 3
**Presentation:** 4
**Contribution:** 3
**Rating:** 6
**Confidence:** 3

**Summary:**

The paper presents SwiftTS, a learning-based framework for selecting pre-trained time series models efficiently. It uses a dual-encoder architecture (data encoder + model encoder) and multi-task meta-learning to predict model–dataset compatibility without exhaustive fine-tuning. Experiments on 14 datasets and 8 models show strong gains over existing methods.

**Strengths:**

- Addresses an important and under-explored problem in time series foundation model selection.
- Well-designed method combining meta, topological, and functional model embeddings.
- Extensive experiments with clear, consistent improvements.

**Weaknesses:**

- The design choices for the data and model encoders appear somewhat heuristic and lack sufficient justification. For example, why does the model encoder capture domain information while the data encoder does not? The paper could be strengthened by clarifying the design rationale of these encoders.
- The meta-learner is trained on a relatively small pool (14 datasets × 8 models); a data-efficiency analysis or a discussion explaining why this scale suffices to learn reliable dataset–model correlations would improve credibility.
- The functional embedding is obtained by probing models with Gaussian noise, which seems heuristic. Random noise may not reflect how models respond to real-world temporal structures; a justification for using Gaussian inputs would be helpful.

**Questions:**

See weaknesses.

---

> ### Author Response · Authors · 2025-11-21
>
> Thank you for your suggestion. We try to address your concerns in the following ways:
>
> Weaknesses:
>
> 1. Since the model encoder already incorporates domain information, injecting similar domain-specific cues into the data encoder would likely lead to superficial matching. For example, UniTS (pre-trained on traffic data) would be assigned a higher predicted score than Moirai on the Traffic dataset simply due to domain overlap, even though Moirai actually performs better. Although it is empirically observed that models often perform better on data similar to their pre-trained domain, this does not always hold. Thus, a selection strategy that over-relies on domain alignment risks capturing spurious correlations rather than genuine model–dataset compatibility.
>
>     In contrast, our patch-based design for the data encoder follows a well-established and effective approach for modeling temporal data. By processing diverse sampled subsets from each dataset and applying mean pooling, it produces compact embeddings that capture shared temporal patterns while remaining robust to sample variability. The resulting data embeddings thus serve as an expressive summary for computing fine-grained compatibility with model embeddings. The patchwise cross-attention visualization in Figure 6 further illustrates that the data encoder effectively captures fine-grained temporal characteristics.
>
>     For the model encoder, directly embedding a model’s full structure is challenging. Instead, we represent the model through three complementary components: meta-information, topological structure, and functionality. This design aligns with human intuition in model selection (e.g., “choose a larger model for complex tasks”), which has been largely overlooked in existing methods. As shown in the ablation study (Table 3), each component contributes to the model selection process, and their combination yields the best overall performance.
>
>     In the revised paper (lines 176-178 and 201-204), we clarify the design rationale of these encoders.
>
>
> 2. Although our model selection pool contains only 14 datasets and 8 models, the SwiftTS is effectively trained on a much richer and more diverse set of dataset-model pairs. On the data side, we sample multiple subsets from each original dataset to form inputs for the data encoder, generating a broad collection of representations that capture diverse characteristics of downstream tasks. On the model side, the pre-trained models are highly heterogeneous in both architectures and training objectives, further enriching the diversity of model choices. Additionally, each dataset-model pair is evaluated under four different forecasting horizons, increasing scenario variability and resulting in a total of 5,408 dataset-model pairs. Finally, our transferable cross-task meta-learning setup simulates out-of-distribution conditions, encouraging the SwiftTS to capture robust, generalizable correlations rather than overfitting to specific tasks.
>
>     Collectively, these designs greatly amplify the effective training signal, enabling our approach to generalize well and suffice to learn reliable dataset–model correlations.
>
> 3. The choice of Gaussian inputs aims to obtain a generic and data-agnostic embedding of the model’s functional characteristics. Although using real or synthetic time-series inputs can reflect how a model responds to the real-world temporal structures, it causes the resulting functional embedding to inadvertently inherit biases or domain-specific priors from the chosen probing data. For example, a model would yield a more favorable embedding than others simply because the probing inputs are similar to its pre-trained data, which is unfair to other models. Moreover, in practice, the downstream tasks and their associated temporal structures are unknown when the SwiftTS is trained. Relying on any specific type of real-world probing data could thus introduce unintended preferences into the model selection process. In contrast, random Gaussian noise serves as a neutral stimulus, allowing us to observe the model’s intrinsic input-output behavior without imposing external assumptions.
>
>     We clarify the design of the probing models with Gaussian noise in the revised paper (lines 253-258).

---

> ### Author Response · Authors · 2025-11-28
> **Looking forward to your feedback**
>
> Dear Reviewer ERT9,
>
> Thank you sincerely for your time and thoughtful feedback on our paper. With the discussion period coming to a close, we would like to confirm whether our responses have adequately addressed your main concerns.
>
> We have carefully considered all your comments and devoted substantial effort to revising the paper accordingly. Should any questions or suggestions remain, we would deeply appreciate the opportunity to address them and provide any further information needed.
>
> If you feel that our responses have resolved your concerns, we would also be thankful if you could take this into consideration when updating your evaluation.
>
> Thank you once again for your constructive feedback and support.
>
> Best regards,
>
> The Authors

---

### Official Review · Reviewer_SQsf · 2025-10-31

**Soundness:** 3
**Presentation:** 3
**Contribution:** 3
**Rating:** 6
**Confidence:** 3

**Summary:**

This paper proposes SwiftTS, a framework for fast and scalable selection of pre-trained time series models via multi-task meta-learning. Rather than fine-tuning each candidate model or relying on computationally expensive feature extraction for each selection, SwiftTS uses a lightweight dual-encoder to embed datasets and models, computes patchwise compatibility via cross-attention, and applies a horizon-adaptive mixture-of-experts approach. Further, the method leverages transferable cross-task meta-learning across datasets and forecasting horizons to enhance out-of-distribution robustness. The framework is evaluated on 14 real-world datasets and 8 pre-trained model families, showing state-of-the-art ranking accuracy and efficiency across a variety of horizons and domains.

**Strengths:**

1. The proposed dual-encoder architecture is well-conceived and technically sound. One encoder incorporates temporal awareness through the use of patching and attention mechanisms, while the other enables knowledge injection by integrating architectural metadata, graph-based topological structures, and functional embeddings derived from model behavior. This design is particularly well-justified for highly diverse model repositories.

2. The manuscript presents extensive experimental results and visualizations, which provide strong and multifaceted evidence supporting the effectiveness of the proposed model.

3. The application of meta-learning to address the heterogeneity of time-series domains and various pretrained models is highly appropriate and demonstrates solid methodological reasoning.

**Weaknesses:**

Although the paper shows runtime savings over fine-tuning, there's insufficient discussion of the the practical scaling beyond a fixed model zoo. For example, how does graph2vec embedding scale with hundreds or thousands of models with complex DAGs? Is there a resource bottleneck for functional embedding inference as the number of candidate models grows? The scalability arguments are more empirical than architectural; a more detailed analysis would be valuable.

**Questions:**

1. How is the meta information embedded and utilized? In particular, how are the five types of meta information combined within the model?

2.SwiftTS performs selection among multiple pretrained models. Could it support the addition or removal of models without retraining SwiftTS? This consideration may have implications for the scalability of the approach.

---

> ### Author Response · Authors · 2025-11-21
>
> Thank you for your suggestion. We try to address your concerns in the following ways:
>
> Weakness:
> 1. The construction of both topological and functional embeddings is an **offline**, dataset-agnostic process. It only requires feeding each candidate model a fixed input with the correct dimensionality (e.g., a fixed set of Gaussian random noise vectors, rather than full downstream datasets). This incurs a total preprocessing cost of $\mathcal{O}(N \cdot T_{emb})$, where $N$ is the number of candidate models and $T_{emb}$ denotes the time required to compute the embeddings for a single model. We empirically evaluate the runtime (in seconds) required to obtain these embeddings for each model:
>
>     | Embedding    | TimesFM  |  UniTS   | Moment | TTM | Moirai |  Rose | Timer |Chronos  |
>     |------------|----------|----------|--------|-------|----------|----------|----------|----------|
>     | functional | 0.714     | 0.335   | 0.578  | 0.247  | 0.606   | 0.438     | 0.396      | 0.683      |
>     | topological| 1.092     | 0.471   | 2.328  | 0.317  | 1.084   | 0.532     | 0.490      | 1.720      |
>
>     In contrast, existing approaches require substantial forward-pass computation over the entire downstream dataset for every candidate model. This results in a total cost of $\mathcal{O}(N \cdot T_{f}(D))$. Here, $T_{f}(D)=\Omega(|D|\cdot C_{model})$, where $|D|$ is size of dataset and $C_{model}$ represents​ the per-sample computational cost. Thus, their runtime depends not only on model complexity but also heavily on the size of the downstream data. For example, on the same hardware, the forward-pass runtime of TimesFM (200M) is 654.98 seconds on the relatively small AQShunyi dataset, but increases to 5,426.12 seconds on the larger Solar dataset.
>
>     Overall, because $T_{emb} \ll T_{f}(D)$ and the embedding computation is performed offline, the linear cost growth with respect to $N$ remains highly tractable. The per-model embedding overhead of SwiftTS is minimal and does not introduce a resource bottleneck as the candidate models increases. Moreover, since embedding computation is fully offline, it can be precomputed if still concerns about runtime.
>
>     In the revised paper, we provide the efficiency analysis of the topological and functional embeddings in Appendix A.6.
>
> Questions:
> 1. The five types of meta-information include: model architecture (category), model capacity (scalar), model complexity (scalar), model dimension (scalar), and pre-trained domain (category).
>
>     - Categorical features are converted into one-hot vectors. For the pre-trained domain, which may involve multiple labels, a multi-label one-hot vector is employed to represent each domain.
>     - Scalar features are normalized based on their minimum and maximum values across all models.
>
>     The resulting normalized scalar features and one-hot encoded categorical features are then concatenated to construct a unified meta-information embedding. Since the meta-information embedding may have a semantic gap with the topological and functional embedding, we apply a linear projection that maps it into a shared latent space. The projected embedding is subsequently integrated with the topological and functional embeddings, as described in Equation (3), to generate the final model embedding used in our model selection framework.
>
>    We have added the embedding and utilization details of meta-information to the revised paper in Appendix A.5.

---

> ### Author Response · Authors · 2025-11-21
>
> Questions:
>
> 2. Since removing models from the original hub does not change the relative ordering of the remaining models, SwiftTS naturally supports model removal without retraining. However, as SwiftTS is trained for scenarios with out-of-distribution data rather than completely new model types, directly evaluating it on newly added models may lead to some performance degradation. This can be mitigated by fine-tuning SwiftTS with additional model–dataset pairs that include the new models. To validate this, we expand the original hub with six additional models (Chronos-mini and Chronos-small, Moirai-small and Moirai-large, and TimeMoE-base and TimeMoE-large) and compare SwiftTS with existing baselines. The table below reports the average weighted Kendall’s $\tau_\omega$ across horizons on 14 datasets:
>
>     |          | RankME  | LogME   | Regression | Etran   | DISCO   | Model spider | zero-shot | SwiftTS |
>     |----------|---------|---------|------------|---------|---------|--------------|-----------|---------|
>     | ETTh1    | -0.207  | 0.103  | 0.074      | 0.207   | 0.106   | 0.236     | $\underline{0.274}$ | **0.402** |
>     | ETTh2    | -0.066  | -0.018  | -0.056     | 0.175   | 0.109   | $\underline{0.347}$ | 0.248     | **0.423** |
>     | ETTm1    | 0.050   | 0.131   | 0.111      | 0.217   | 0.065   | 0.149     | $\underline{0.290}$     | **0.415** |
>     | ETTm2    | -0.029  | 0.022   | 0.075      | 0.303   | 0.208   | 0.151     | $\underline{0.304}$     | **0.453** |
>     | Electricity | 0.174 | -0.019  | 0.132      | 0.100   | 0.043   | 0.089    | $\underline{0.188}$     | **0.383** |
>     | Traffic  | 0.069   | 0.081   | 0.086      | 0.065   | $\underline{0.341}$   | 0.175     | 0.190     | **0.453** |
>     | Solar    | 0.005   | 0.190   | 0.105      | 0.189   | 0.090   | **0.243** | 0.139     |$\underline{0.240}$   |
>     | Weather  | 0.121   | 0.080   | 0.032      | **0.251** | 0.227   | 0.173   | 0.209     | $\underline{0.238}$   |
>     | Exchange | 0.032   | 0.019   | 0.151      | 0.129   | $\underline{0.272}$  | 0.095     | 0.024     | **0.286** |
>     | ZafNoo   | -0.150  | -0.020  | 0.087      | -0.016  | 0.092   | $\underline{0.440}$     | 0.186     | **0.649** |
>     | CzeLan   | -0.257  | -0.076  | -0.110     | 0.150   | 0.047   | $\underline{0.333}$     | 0.219     | **0.515** |
>     | AQShunyi | -0.077  | 0.004   | -0.012     | 0.021   | 0.269   | $\underline{0.455}$ | 0.438     | **0.575**   |
>     | Wind     | -0.242  | 0.176   | 0.189      | $\underline{0.228}$ | 0.039   | 0.181   | 0.056     | **0.310** |
>     | PEMS08   | 0.112   | 0.037   | 0.009      | 0.092   | 0.097   | 0.267     | $\underline{0.302}$ | **0.373**   |
>
>     The results indicate that enlarging and diversifying the model pool makes the selection task more challenging, causing some degradation across all methods. Nevertheless, SwiftTS is considerably less sensitive to this increased heterogeneity and continues to outperform existing approaches, showing its scalability.
>
>     We incorporate the scalability analysis of the model hub in Appendix A.7 of the revised paper.

---

> ### Author Response · Authors · 2025-11-28
> **Looking forward to your feedback**
>
> Dear Reviewer SQsf,
>
> Thank you sincerely for your time and thoughtful feedback on our paper. With the discussion period coming to a close, we would like to confirm whether our responses have adequately addressed your main concerns.
>
> We have carefully considered all your comments and devoted substantial effort to revising the paper accordingly. Should any questions or suggestions remain, we would deeply appreciate the opportunity to address them and provide any further information needed.
>
> If you feel that our responses have resolved your concerns, we would also be thankful if you could take this into consideration when updating your evaluation.
>
> Thank you once again for your constructive feedback and support.
>
> Best regards,
>
> The Authors

---

### Author Response · Authors · 2025-12-01
**Summary of rebuttals & revisions**

Thank you very much for taking the time to review this paper. Our paper proposes **SwiftTS**, a novel framework for **fast and scalable selection** of pre-trained time series models via multi-task meta-learning.

The reviewers provided positive feedback on our work, describing the proposed SwiftTS as **"addressing a practically important yet underexplored challenge"**, with **"solid methodological reasoning"**  and being **"well-conceived and technically sound"**. The experimental evaluation was praised as **"extensive experiments with clear, consistent improvements"**, providing **"strong and multifaceted evidence"**. The presentation was also commended as **"excellent"** and **"good"**.

We sincerely thank all the reviewers for their insightful comments, and have made every effort to address all the concerns and revise the paper accordingly:
- Methodology:
    - **Rationale for probing models with Gaussian noise (Reviewer ERT9 W3 *&* SnbV W7)** : We clarify that random Gaussian noise serves as a neutral stimulus that reveals the model’s intrinsic input-output behavior, avoiding the domain-specific biases introduced by real or synthetic time-series inputs.

        $\text{Revised}:$ $\underline{\text{Clarification of Gaussian noise (lines 253-258).}}$
    * **Details of constructing meta-training tasks (Reviewer SnbV W9 *&* W10)** : Our cross-dataset and cross-horizon sampling strategy ensures the support and query sets come from disjoint datasets and horizons, thereby avoiding distribution leakage and task redundancy.

        $\text{Revised}:$ $\underline{\text{Meta-training task construction details (lines 321-331).}}$
    * **Variance control in multiple-subset sampling (Reviewer SnbV W2)** : The sampled subsets are fixed once the meta-dataset is constructed, ensuring consistent encoder inputs across runs. Figures 6(a) and (b) show similar attention weights across subsets, further confirming that the meta-dataset is insensitive to sampling randomness.

        $\text{Revised}:$ $\underline{\text{Clarification of sampling variance (lines 190-198 and 449-451).}}$
    - **Why SwiftTS suffices to learn dataset-model correlations (Reviewer ERT9 W2)** : It leverages diverse dataset representations (via multi-subset sampling), a heterogeneous model pool, multi-horizon evaluations, and a transferable cross-task meta-learning that simulates OOD scenarios.
- Experiments:
    - **Scalability of larger, more heterogeneous model pools (Reviewer SQsf Q2 *&* Reviewer SnbV W8)** : We extend evaluation to 14 diverse pre-trained models (including 6 newly added) across 14 datasets, showing SwiftTS remains robust as the model pool grows.

        $\text{Newly added}:$ $\underline{\text{Appendix A.7: Scalability of the model hub.}}$
    - **Efficiency of topological and functional embeddings (Reviewer SQsf W1 *&* Reviewer SnbV W1)** : We add theoretical time-complexity analysis and empirical runtime comparisons, showing that per-model embedding cost of SwiftTS is minimal and can be fully precomputed offline.

        $\text{Newly added}:$ $\underline{\text{Appendix A.6: Efficiency Analysis of topological and functional embeddings}}.$
    - **Actual performance of Top1-selected models (Reviewer SnbV W4)** : We report the forecasting performance of the top-1 model selected by each method, showing that SwiftTS not only selects effectively but also delivers superior actual performance across datasets.

        $\text{Newly added}:$ $\underline{\text{Appendix A.8: The perforamnce of Top1-selected model}.}$
    * **Expanded comparisons with Autoforecast (Reviewer SnbV W6)** : We add AutoForecast as a new meta-learning baseline, providing 56 new results in Table 1 to further validate SwiftTS’s effectiveness.

        $\text{Newly added}:$ $\underline{\text{Table 1: Method comparison across 14 datasets}.}$
    - **Ground-truth fine-tuning results of the pre-trained models (Reviewer SnbV W5)** : We add these results to ensure fairness and reproducibility.

        $\text{Newly added}:$ $\underline{\text{Appendix A.9: Ground-truth Fine-tuning Results}.}$

- Additional Clarifications:
    - **Reviewer SnbV W3**: $\underline{\text{Semantic meanings of topological structure (revised in lines 240–245).}}$
    - **Reviewer SQsf Q1**: $\underline{\text{Details of Meta-information of Pre-trained Models (revised in lines 332-342).}}$
    - **Reviewer ERT9 W1**: $\underline{\text{Details of dual-encoder design (revised in lines 176-178 and 201-204).}}$

The reviewers’ insightful suggestions have greatly strengthened our paper. **In response, we have conducted an extensive revision and added over 400 new experimental results to address the above concerns.** All revisions are highlighted in $\underline{\text{blue}}$.

We hope the clarification above can help you better evaluate our paper. Thank you once again for your time and thoughtful consideration.

---

### Meta-Review · Area_Chair_eppL · 2026-01-03

**Summary:**

This paper proposes SwiftTS, a novel framework for fast and scalable selection of pre-trained time series models via multi-task meta-learning. Most reviews are positive. Specifically, the reviewers describe the proposed SwiftTS as "addressing a practically important yet underexplored challenge", with "solid methodological reasoning" and being "well-conceived and technically sound". The experimental evaluation was praised as "extensive experiments with clear, consistent improvements", providing "strong and multifaceted evidence". The authors' rebuttal has adeptly addressed the majority of concerns raised by the reviewers.
Overall, I recommend the acceptance of this submission. Additionally, I expect that the authors will incorporate the new results and suggested modifications from the rebuttal phase into the final version.

**Reviewer Concerns:**

Most concerns have been well solved. Specifically, most concerns  below have been well answered:
1) Rationale for probing models with Gaussian noise (Reviewer ERT9 W3 & SnbV W7)
2) Details of constructing meta-training tasks (Reviewer SnbV W9 & W10)
3) Variance control in multiple-subset sampling (Reviewer SnbV W2)
4) Why SwiftTS suffices to learn dataset-model correlations (Reviewer ERT9 W2)
5) Scalability of larger, more heterogeneous model pools (Reviewer SQsf Q2 & Reviewer SnbV W8)
6) Efficiency of topological and functional embeddings (Reviewer SQsf W1 & Reviewer SnbV W1)
7)Actual performance of Top1-selected models (Reviewer SnbV W4)

**Reviewer Scores:**

SQsf may keep the original score as 6.
ERT9 may keep the original score as 6.
SnbV may raise the score from 2 to 6.

---

### Decision · Program_Chairs · 2026-01-26

Accept (Poster)